# Triton: Topography and Geology of a Probable Ocean World with Comparison to Pluto and Charon

Paul M. Schenk [1,*], Chloe B. Beddingfield [2,3], Tanguy Bertrand [3], Carver Bierson [4], Ross Beyer [2,3], Veronica J. Bray [5], Dale Cruikshank [3], William M. Grundy [6], Candice Hansen [7], Jason Hofgartner [8], Emily Martin [9], William B. McKinnon [10], Jeffrey M. Moore [3], Stuart Robbins [11], Kirby D. Runyon [12], Kelsi N. Singer [11], John Spencer [11], S. Alan Stern [11] and Ted Stryk [13]

1   Lunar and Planetary Institute, Houston, TX 77058, USA
2   SETI Institute, Palo Alto, CA 94020, USA; chloe.b.beddingfield@nasa.gov (C.B.B.); rbeyer@seti.org (R.B.)
3   NASA Ames Research Center, Moffett Field, CA 94035, USA; tanguy.bertrand@nasa.gov (T.B.);
    dale.p.cruikshank@nasa.gov (D.C.); jmoore@mail.arc.nasa.gov (J.M.M.)
4   School of Earth and Space Exploration, Arizona State University, Tempe, AZ 85202, USA; CBierson@asu.edu
5   Lunar and Planetary Laboratory, University of Arizona, Tucson, AZ 85641, USA; vjbray@lpl.arizona.edu
6   Lowell Observatory, Flagstaff, AZ 86001, USA; grundy@lowell.edu
7   Planetary Science Institute, Tucson, AZ 85704, USA; cjhansen@psi.edu
8   Jet Propulsion Laboratory, Pasadena, CA 91001, USA; jason.d.hofgartner@jpl.nasa.gov
9   National Air & Space Museum, Washington, DC 20001, USA; martines@si.edu
10  Department of Earth and Planetary Sciences, Washington University in Saint Louis,
    Saint Louis, MO 63101, USA; wbmckinnon@wustl.edu
11  Southwest Research Institute, Boulder, CO 80301, USA; stuart@boulder.swri.edu (S.R.);
    ksinger@boulder.swri.edu (K.N.S.); spencer@boulder.swri.edu (J.S.); alan@boulder.swri.edu (S.A.S.)
12  Johns Hopkins Applied Physics Laboratory, Laurel, MD 20707, USA; kirby.runyon@jhuapl.edu
13  Humanities Division, Roane State Community College, Harriman, TN 37748, USA; strykt@roanestate.edu
*   Correspondence: schenk@lpi.usra.edu

**Abstract:** The topography of Neptune's large icy moon Triton could reveal important clues to its internal evolution, but has been difficult to determine. New global digital color maps for Triton have been produced as well as topographic data for <40% of the surface using stereogrammetry and photoclinometry. Triton is most likely a captured Kuiper Belt dwarf planet, similar though slightly larger in size and density to Pluto, and a likely ocean moon that exhibited plume activity during Voyager 2's visit in 1989. No surface features or regional deviations of greater than ±1 km amplitude are found. Volatile ices in the southern terrains may take the form of extended lobate deposits 300–500 km across as well as dispersed bright materials that appear to embay local topography. Limb hazes may correlate with these deposits, indicating possible surface–atmosphere exchange. Triton's topography contrasts with high relief up to 6 km observed by New Horizons on Pluto. Low relief of (cryo)volcanic features on Triton contrasts with high-standing massifs on Pluto, implying different viscosity materials. Solid-state convection occurs on both and at similar horizontal scales but in very different materials. Triton's low relief is consistent with evolution of an ice shell subjected to high heat flow levels and may strengthen the case of an internal ocean on this active body.

**Keywords:** Triton; ocean world; Neptune; topography; cartography; Pluto

## 1. Introduction

Neptune's large satellite Triton (Figure 1) was revealed by Voyager 2 in 1989 to be a diverse, youthful, and active body, and is now a major focus of planetary exploration [1,2] and has been designated as the top priority target for future exploration of icy ocean worlds (e.g., [3,4]). Triton's very young surface (perhaps as young at ~10 myr; [5–8]), outer ice-rich shell, geologic complexity [6,9], and active atmospheric plumes [10] related to either solar or geothermal heating [11,12] all suggest that Triton, most likely a captured Kuiper belt dwarf

planet, has been subject to high levels of internal heat, may be currently active, and likely has an internal ocean today (e.g., [3,4,7,13,14]). The known surface ices on Triton include water, methane, CO and $CO_2$ ices [15–17]. Unlike Cassini or New Horizons, Voyager did not carry a mapping infrared spectrometer and did not map out the distribution or geologic correlations of these materials. These factors also hint that Triton may have implications for the nature, diversity, and likelihood of habitable environments across the Galaxy.

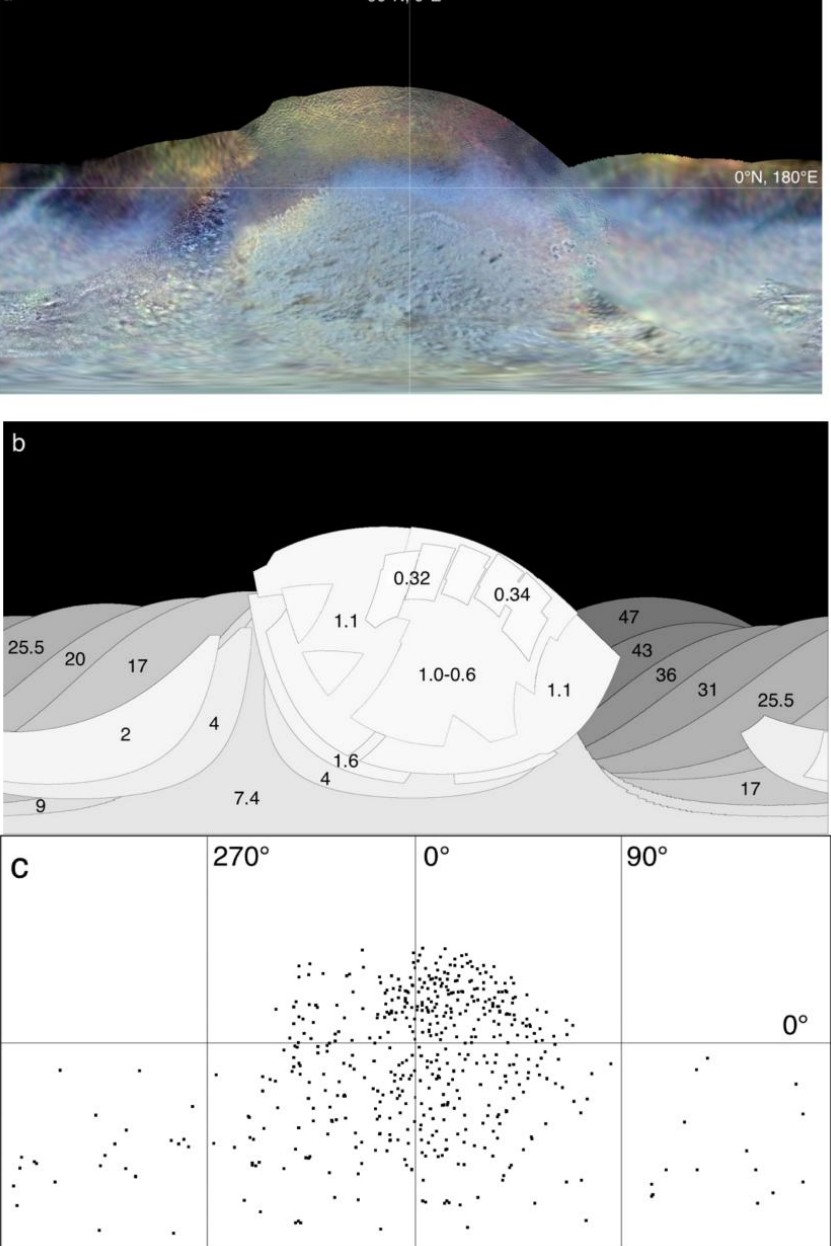

**Figure 1.** (**a**) Global color map mosaic of Triton. Color-composite uses orange, green, and UV filters for red, green, and blue colors. Simple cylindrical projection at 0.35 km/pixel from −180° to 180° E longitude (same map projection used in all similar maps unless noted). (**b**) Resolution map of Triton showing footprints and effective image resolutions in km/pixel for the observations used in construction of the global map in Figure 1a. (**c**) Map of match point distribution used for updated global control network for Triton and global mosaic shown in Figure 1a,b.

As for other large or active icy worlds, a key constraint on our understanding of the nature of Triton's ice shell and the possible ocean below (with implications for its

exploration) is surface topography, which reflects the geophysical processes deforming it and influences how the shell can be probed with radar and other instrumentation. Here, we describe the distribution, quality, and implications of newly derived stereo and shading derived topographic data for Triton, together with new global multicolor surface maps produced during the topography derivation. The topography mapping is necessarily very incomplete, limited to equatorial and near northern latitudes, and sparse measurements in the southern hemisphere. These distributed data are not useful for global studies of shape (due to peculiar aspects of the Voyager imaging system and coverage) but provide our only insights into the topographic characteristics of the known terrain types. We take advantage of these new data to offer insights into and some speculations about the observed features in anticipation of future mapping missions to Triton and the Neptune system. Finally, we contrast Triton with the most closely related (dwarf) planet that has been explored at close range, Pluto, which may also be or have been an ocean world (e.g., [18]), and its satellite Charon.

## 2. Materials and Methods

The only resolved imaging data of Triton are from Voyager 2 in 1989 [5]. The Voyager imaging of Triton included resolved images with 4 to 0.34 km pixel scales of ~40% of Triton's surface observed near the time of closest approach (Figure 1), which are factors of 3 or so poorer than those acquired by New Horizons at Pluto.

Aside from sequential approach imaging at ~47–8 km/pix (which shows albedo patterns but do not resolve geologic features), Voyager 2 acquired five imaging observations of relevance to production of a global map (Figure 1). The first resolved imaging was a set of 6 filter images of the whole disk acquired at 4 km/pix and a phase angle of 39° (henceforth VTLON; Figure 2a). This was followed by a partial hemispheric multi-frame mosaic (henceforth VTCOLOR; Figure 2b) at 1.64–1.45 km/pix and phase angles of ~62° acquired in 3 color filters (UV, Violet, and Green with some of the longer exposure frames are smeared by a few pixels) and a clear filter. A second full hemispheric, 17-frame mosaic in a clear filter (henceforth VTMAP; Figure 2c) was acquired at 1.3–0.6 km/pix and phase angles of 63–79°. A 10-frame mosaic was acquired near the time of closest approach at 0.42–0.34 km/pix and phase angles of 97–116° (henceforth VTERM; Figure 2d). Voyager 2 also obtained five overlapping images of Triton on departure of an area centered near 35°S, 250°E that was poorly resolved on approach (Figure 2e). These crescent images are at pixel scales of ~2 to ~4 km, and at phase angles of 135° (henceforth VTCRSCNT) and are included in the global mosaic (Figure 1). While quasi–circular bright features of comparable size to cantaloupe cells and the dark quasi–circular features within the southern terrains (see below) are evident in the crescent imaging, it is not obvious from these low-exposure images whether the bright features correspond with any of these structures. These crescent images also reveal elongated dark streaks suggestive of low clouds or plume trails [11], which occur at similar latitudes as the plumes observed in the mapping mosaics.

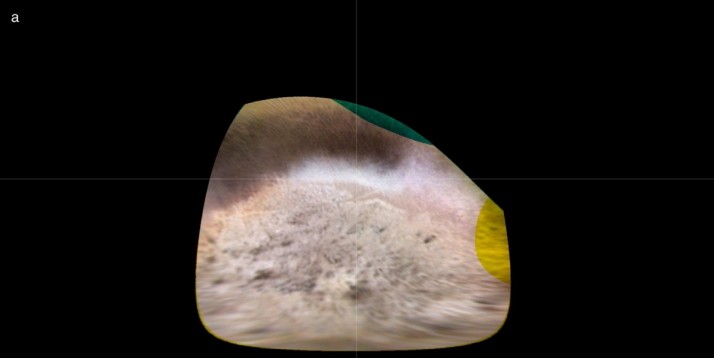

**Figure 2.** *Cont.*

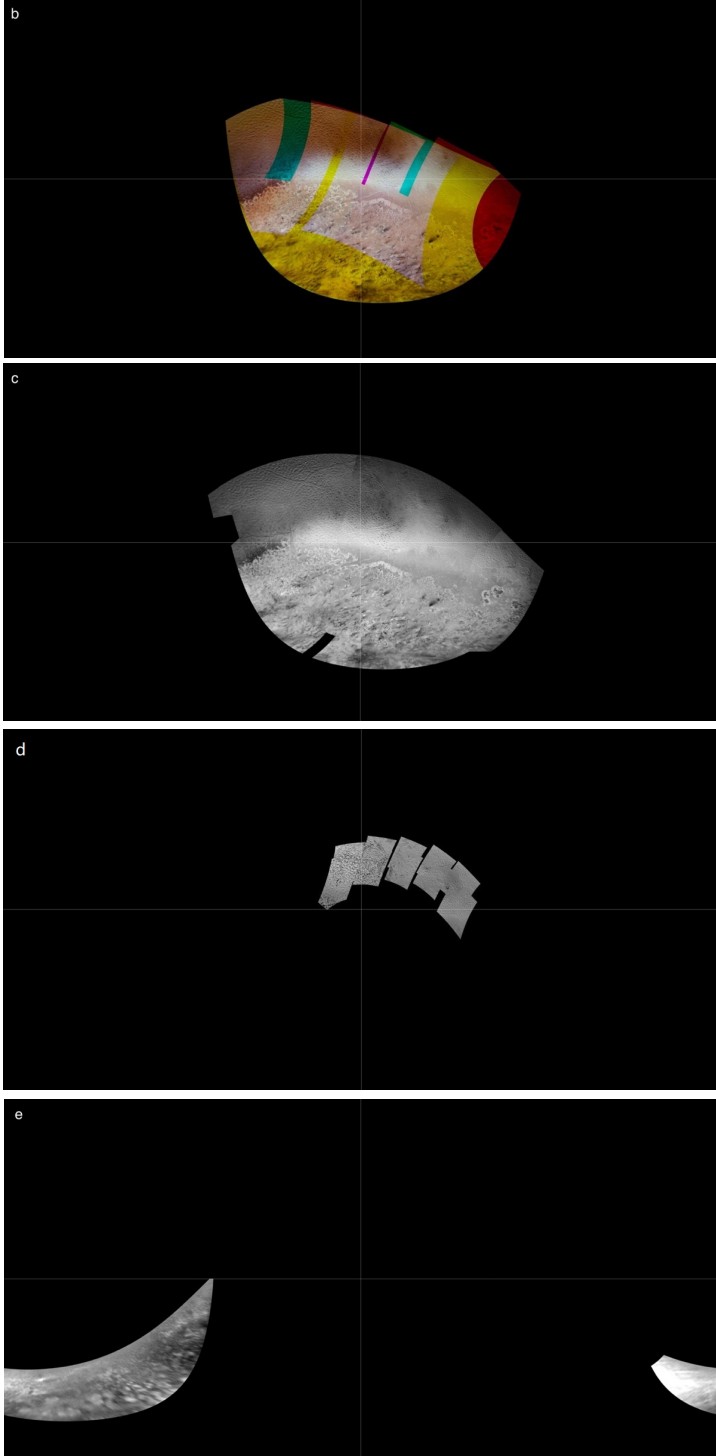

**Figure 2.** (**a**) Five color mapping mosaic of Triton at 4 km pixel scales and ~40° phase angles. This color composite shows the green, violet, and UV filter images as red, green and blue colors. Image ID c1138639.imq through c1138715.img. (**b**) Partial three-color mapping mosaic of Triton at 1.6 to 1.4 km pixel scales and ~61° phase angles. This color composite shows the green, violet, and UV filter images as red, green and blue colors. Image ID c1139255.imq through c1139323.img. (**c**) Mapping mosaic of Triton in clear filter at 1.0 to 0.6 km pixel scales and 63–79° phase angles. Image ID c1139340.imq to c1139533.img. (**d**) Partial mapping mosaic of Triton in clear filters at 0.42 to 0.34 km pixel scales and 98–116° phase angles. Image ID c1139607.imq to c1139629.imq. (**e**) High phase angle image of Triton at 2–4 km pixel scales and ~135° phase angles. Image ID c1140959.imq, c1140138.imq and c11340614-c1140632.imq.

### 2.1. Geometric Registration and Control Network

The Voyager and New Horizons flyby characteristics in 1989 and 2015 (rapid approaches to slowly rotating bodies) were very similar for Triton and Pluto. This resulted in global mapping products that were also very similar (low resolution for the approach hemisphere, much higher on the closest approach (CA) hemisphere) with the exception that the Pluto observations were at factors of 2–4 better pixel scales than at Triton. Solar obliquity was also very similar during the two encounters, resulting in a non-illuminated region poleward of ~40° on both bodies. The high phase angle images acquired at 2 to 4 km/pix on departure allow us to examine part of the region between 160° and 260° E longitude viewed at low resolution on approach, albeit obliquely. As Triton apparently has such low relief (e.g., [6]), these lower-resolution images reveal apparent brightness patterns but do not reveal shadows or prominent topography. These images were acquired at phase angles of ~137°, in contrast to the Pluto departure crescent images at ~165°, which are even more difficult to interpret. Another difference is that the approach images at Pluto were at ~15° phase angle but those of Triton were at ~27°, though this made little practical difference as the Triton approach images are too low in resolution and Triton too low in relief for these images to be interpretable geologically.

As a result of the general similarities of the Triton and Pluto datasets, we use the same mapping and geometric registration techniques as used for the global mapping of Pluto and Charon [19,20]. The reader is referred to those works for descriptions of the basic process. Here we describe aspects of the process related to the distinct characteristics of the Voyager imaging system and the lower resolutions of the Triton datasets.

To produce the updated image mosaic of Triton (Figure 1) with the best quality image at each point, and optimal derived topographic map products (see next section), an update of the camera pointing vectors for the Voyager images was required. As described in the Pluto/Charon reports [19,20], we use the USGS-ISIS "jigsaw" bundle adjustment algorithm to update camera pointing vectors for the Triton images simultaneously based on match points linking features in the images. Our effort (Figure 1c) represents an update to the revised map posted online [21] and differs from other recent efforts to update the Triton network [22] in that we include all color images, the two highest resolution wide angle images, the resolved departure crescent images, and all approach images from the final Triton rotation prior to the closest approach.

The most relevant concern in control network solutions for Triton was the factor of up to ~50 difference in resolution of the distant approach images on the sub-Neptune side, which made identification and correlation of distinctive recognizable match points in the overlapping high-resolution imaging more difficult. We adopted the approach used at Pluto [19] in which an initial set of points was identified, the bundle adjustment run, and maps made for each image. Blink comparison between overlapping maps showed where misalignments persisted and points in that region were revised, deleted, or supplemented based on which points were reported to have the highest residuals from the bundle adjustment. The process was repeated until misalignments were judged negligible. It should be recognized, however, that the combination of the potential Voyager vidicon distortion uncertainties (described below) and the very low resolution of the approach images will result in maps of Triton with positional uncertainties of roughly 1–2 km in the best resolved areas and as much as ~10 km in lower resolution areas.

### 2.2. Deriving Topography of Triton

Examination of the Voyager 2 images and map mosaics of the observed surface of the sub-Neptune hemisphere of Triton (Figure 1) reveals no long shadows or regional shading variations that could be interpreted (as on Pluto (e.g., [19])) as deep basins, high mountain(s) or broad domical or depressed undulations at resolved scales. Despite this and the importance of Triton as a likely ocean world [4], topographic mapping has been limited [6,23] (Figure 3).

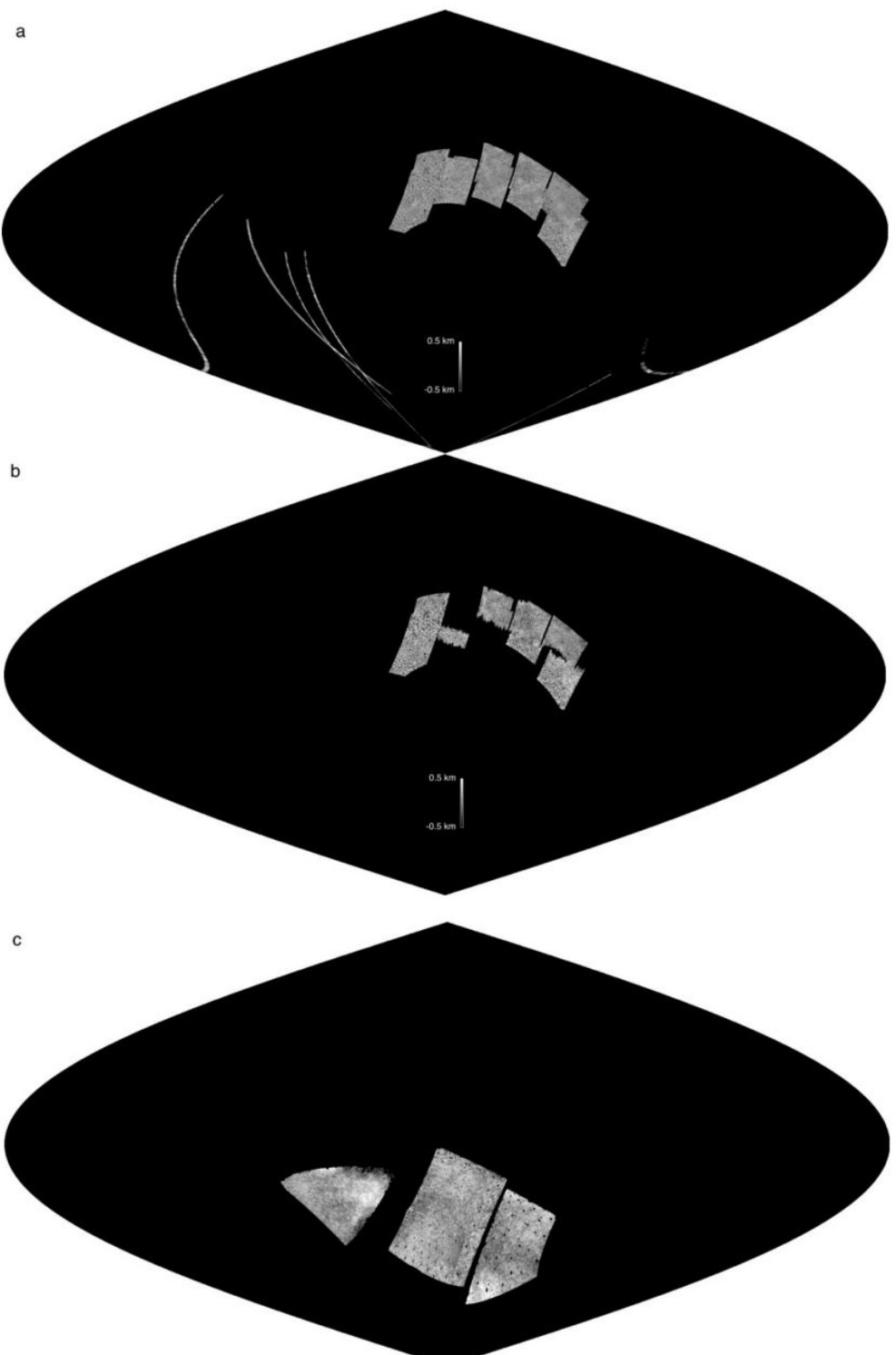

**Figure 3.** (**a**) Global sinusoidal maps showing distribution of stereogrammetric (SG) DEMs for Triton and curvilinear limb profile tracks provided by [23]. Only those stereo DEMs where geologic features are nominally resolved are shown. Map area also corresponds to VTERM CA mosaic outline and coincident PC-DEM. Map limits from −180 to 180° E longitude. (**b**) The figure is same as Figure 3a, except limb tracks are not shown and with those parts of SG-DEM affected by truncated reseaux marks and potential uncertainties in geometric correction masked out. The areas shown are, in principle, not affected by reseaux truncations. (**c**) SG-DEMs of southern hemisphere terrains. Undulations are not regarded as reliable, due to low stereo sensitivity of the input stereo pairs.

For extended topographic mapping, we use well-tested stereogrammetric (SG) and photoclinometric (PC) techniques on Voyager narrow-angle camera imaging data to produce topographic maps over more extended areas within the higher-resolution mapping hemisphere. The SG and PC techniques used here are identical to those described in detail in a series of peer-reviewed works (e.g., [19,20,24,25]) and readers are referred to these works for more detailed descriptions of methods. The Voyager imaging experiment and compression algorithms used at Triton each present distinct challenges, however, that must be described in detail in order to understand the limits of the derived topographic data and uncertainties associated with topographic measurements (see Appendix A).

### 2.3. Limb Profiles and Match Point Radii

Although spatially limited, two techniques provide supplemental topographic coverage supplemental to the main stereogrammetry and photoclinometry mapping. Five published limb profile tracks [23] covering <0.01% of the surface provide limited topographic insights but lack geologic context as they are located in poorly resolved regions of the global map (Figures 3 and 4). These are derived from analysis of relief viewed in projection along the limb. As such, extreme lows in topography may be hidden by foreground relief and prominent highs (including hazes or clouds) may be seen in projection even when not located on the nominal limb track.

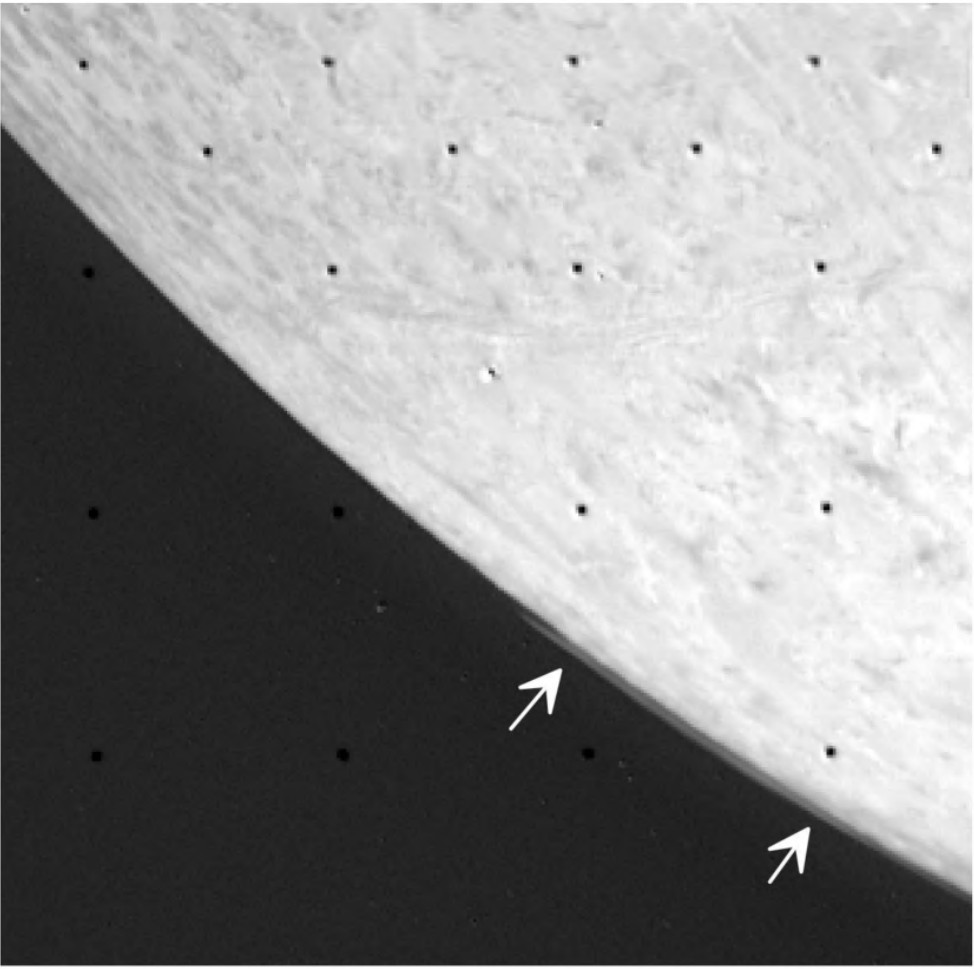

**Figure 4.** Limb image of Triton showing a detached haze layer (arrows) at least 5–7 km above the surface. Image number c1139427.imq. Nominal pixel scale of 1 km, haze centered at ~30° S, 300° E. Surface is brightened to enhance lack of limb relief. Area dominated by cantaloupe terrain and several linear ridges. South is to bottom.

The coordinates of the match points used to update the camera pointing vectors of each image (as described in the preceding section) can also be solved for the radius of the feature from planet center. These control network (CN) match point radii solutions supplement the SG and PC techniques where they do not work well, particularly in the southern areas, as described below. None of these techniques work in the hemisphere opposite to that observed in high resolution due to the inherently poorer quality of those approach images.

*2.4. Stereogrammetry*

Digital stereogrammetry derives topography from observed parallax of surface features but uses scene recognition procedures which sample topography over multiple pixel footprints. Digital stereogrammetry thus provided the most extensive topographic coverage where long-wavelength information is reliable. Heights of features are determined from this measured parallax from standard parallax equations [26] using the derived camera, spacecraft and point vectors. The known atmospheric plumes were observed in Voyager stereo images (Soderblom et al., 1990; Hansen et al., 1990), which provide several possible stereo combinations for Triton on the encounter (Neptune-facing) hemisphere. (Resolutions in the anti-Neptune hemisphere and phase angles are too low to resolve surface features or topography.) The best stereo combination includes the 10-frame VTERM CA mosaic at 0.42–0.34 km/pix (Figures 1b and 2d) and mostly north of the equator, with the 17-frame VTMAP hemispheric mosaic of Triton at pixel scales of 1.3–0.6 km (Figure 3c). The two mosaics provide useful stereo potential and result in SG-DEMs with nominal vertical precisions (or detection thresholds) of 200–350 m in the areas covered in the mosaic. The 10 frames of the VTERM mosaic do not all overlap (Figure 3c), however, and it is not possible to construct an integrated DEM across the entire mosaic that could be used to search for topographic variations along its length.

Triton itself did not 'cooperate' in that it is one of the lowest-relief icy objects visited by either Voyager, as will be shown below. As vertical precision in stereogrammetry is a function of distance and stereo angle between the two observations, such that if terrains are lower in amplitude than the precision of the observation, then larger stereo separation angles, higher resolution imaging, or both would be required to resolve most geologic features stereoscopically. Such was the case for almost all terrains south of the equator observed by Voyager, although these observations were more than sufficient to resolve the 8-km high plumes in the mid-southern latitudes [10], which used Stereo combinations between frames acquired early and late in the global mosaic sequence (Figure 3c). These cover an addition ~10% of the surface, although none of these combinations have vertical precisions of better than ~1400 m and none unambiguously resolve the heights of individual geologic features for single stereo pairs.

There are characteristics of the Voyager Triton stereo datasets that should be understood prior to interpretation of topographic data derived from these images. The Voyager images were calibrated and registered in the USGS ISIS image processing system using the ISIS-standard processing pipeline for Voyager data (available at USGS online). The key difference between the Voyager optical cameras and CCD and other imagers on later missions (Galileo, Cassini, New Horizons) is that they are vidicon cameras in which the image plate was scanned by an electron beam for readout. To correct for geometric distortion in the image plane, ~220 spots (or "reseaux") roughly $5 \times 5$ pixels in size were etched into the plate (Figure 5) and these are mapped out later during image calibration for correction of image distortion. Errors in camera distortion correction resulting from reseaux location errors will result in errors in subsequent geographic placement of pixels on the planet's surface that will be incorrectly interpreted as real parallax displacements and hence real topographic values.



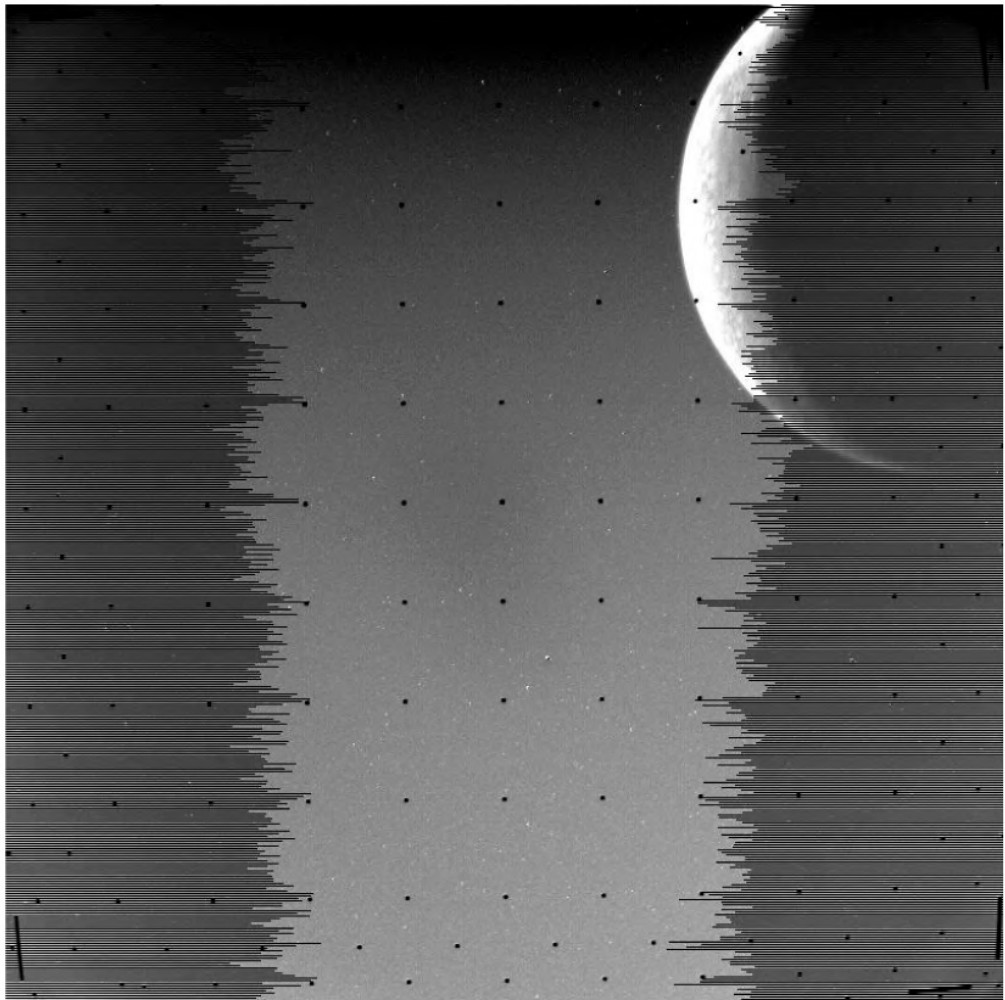

**Figure 5.** Reseaux marks (dark spots) and line dropouts in compressed Voyager images of Triton. Image ID c1140959.imq. Reseaux marks occur in all images.

Correction of geometric distortion in the image plane is precisely known in CCD images and other modern planetary framing imaging systems. On Voyager, distortions in the image readout are dependent on a variety of environmental and scene-dependent characteristics, including the brightness of the object in view, and vary from image to image. Hence, the reseaux locations can change by >1 pixel and must be mapped out in each image before the distortion correction is applied. This distortion is well characterized in the central part of Voyager images but less precisely in the outer ~50 pixels of the image, where geometric distortions are sometimes revealed as topographic distortions. These outermost ~50 pixels should be masked out if overlapping data with less distortion are available. It is understood that, except for these outermost pixels, the Voyager geometric camera distortions are correctable to a fraction of a pixel in the ISIS system.

For the Voyager stereo images at Triton, an additional complication arises. The VTMAP images (and some of the crescent imaging) were acquired in a compression mode that resulted in alternate lines being truncated along the left and right sides of the images (Figure 5). Between 25 and 35% of each such image was affected in this way. Unfortunately, this compression mode also removed half of each reseaux along left and right sides of these images, and as the ISIS reseaux mapping procedures rely on scene-recognition of each reseaux to precisely map its line and sample location, this potentially corrupts reseaux pattern mapping. The derived reseaux positions (and hence derived topography which is based on camera distortion correction) depended on the line interpolation method used. We tested two different methods for interpolation of the missing line sections and found that

images calibrated using USGS ISIS FILLGAP interpolate scheme with cubic interpolation produced DEMs with the least amount of long-wavelength distortions across the scene. In Figure 3b, we mask off those portions of the image where this compression results in missing lines and greater uncertainty in reseaux positioning. This process showed that ~20% of the high-resolution SG-DEM was potentially affected by the line dropouts (Figure 3b). Any residual corruption in reseaux locations and distortions in the dropout areas will likely be in random directions, implying that systematic errors are unlikely, and the maximum relief across these DEM is still very low, indicating that residual distortions are subtle. These dropouts did not occur in any of the approach, VTCOLOR or highest resolution VTERM images, which were compressed using a different algorithm.

### 2.5. Match Point Radii Solutions

The SG and PC mapping was supplemented by radii solutions derived from the control network solution from the ISIS control network bundle adjustment, where the surface radius for each match point connecting multiple images is estimated during the least-squares adjustments of the camera vectors described above. Two-image match point solutions can provide radii estimates but greater vertical precision and confidence can be achieved if multiple stereo images of the same region are used. Such is the case for images c1139503, c1139427, and c113433.imq, which are the last views of the southern hemisphere Voyager acquired and provide the highest parallax with the early images in the sequence. This effort, described in the Southern Terrains section below, is complicated by the fact that many features in southern latitudes are diffuse or unsharp at Voyager resolutions.

### 2.6. Photoclinometry

Photoclinometry provides topographic information at the pixel scale based on shading variations of surface slopes, but is subject to long-wavelength uncertainties due to inherent assumptions about photometric behavior of that surface. The primary stereo combination between the VTMAP and VTERM mapping sequences covers ~5% of the total surface, including substantial areas of the terminator region where solar shading provides important topographic information. The topography of terminator areas can be more completely mapped using shape-from-shading, or photoclinometry (PC; Figure 6). These areas are again limited to terrains (mostly) north of the equator and provide no useful topographic data for Triton's southern hemisphere terrains, but do cover ~8% of the total surface, overlapping with the SG-DEM.

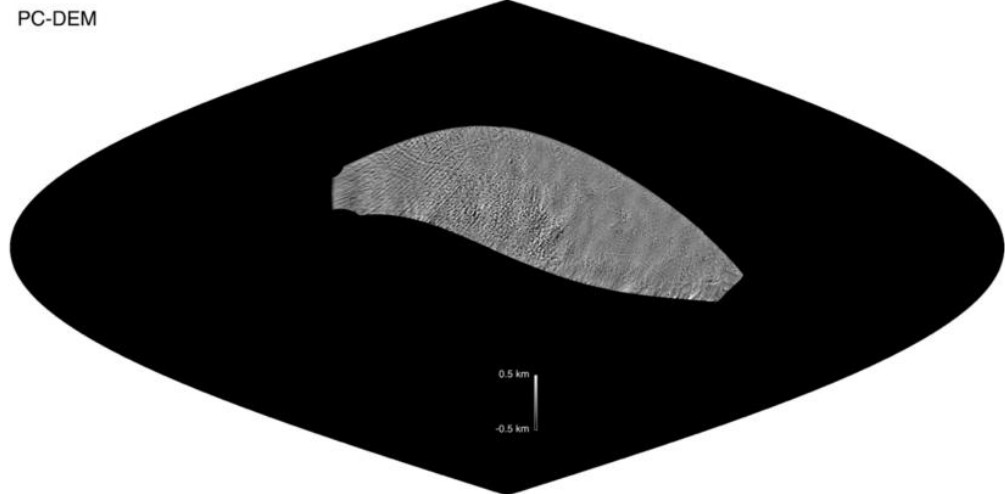

**Figure 6.** Global sinusoidal map showing distribution of photoclinometric (PC) DEM for Triton derived from VTMAP mosaic at 0.9–0.6 km/pix. See Figure 3 for SG-DEMs and Figure 8 for merged SG-PC-DEM tied to CA-mosaic.

The photoclinometry (PC) technique uses the apparent brightness of each pixel to estimate surface slope, making several assumptions regarding the photometric behavior of the surface. This requires that the photometric behavior of the surface relative to viewing geometry be well understood. In photoclinometry, the precision of the camera distortion is relevant only to the placement of each pixel on the surface and does not have any influence on the actual calculation of slope as long as the image is correctly located and the solar illumination is correctly calculated. For Triton, as on other bodies (e.g., [25,27]), we use the so-called lunar-lambertian photometric function [28], which follows the simplified empirical form:

$$DN = A\left[2*L*\cos(i)/(\cos(i) + \cos(e)) + (1 - L)\cos(i)\right] \tag{1}$$

where $e$ and $i$ are the emission and incidence angles, A is the "normal reflectance," and $L$ is the photometric factor related to the Lambertian contribution, which varies with phase angle ($a$) and may vary with terrain type. To use our PC technique, we must estimate $L(a)$ for Triton in the areas of interest. The photometric behavior of Triton is known to be complex [29]. Our approach is to estimate $L(a)$ values for the major brightness units by least squares fitting of Equation (1) to global and regional scale images at a variety of phase angles, including the phase angle ranges of interest for PC mapping. This fit gives a functional form of $L(a) = 0.88 - 0.014(a) + 0.00005(a)$.

Our results show that PC mapping on Triton (and other icy surfaces) is most reliable on image data acquired within ~30° of the terminator and we limit our mapping to these areas, which cover ~8% of the total surface (Figure 6). These are areas where shading due to slope rather than inherent albedo variations of surface materials is the dominant contributor to the apparent brightness of terrains. Variations in photometric properties of the surface and the lack of a true albedo model for all of Triton at kilometer scales can result in the dominance of albedo-related slope artifacts at incidence angles below ~60°. Our PC implementation depends on the assumption that the intrinsic albedo of the surface is uniform, which is often not the case. We use lower resolution images as improvised albedo models of the surface to improve slope estimates and reduce distortions. Even then local albedo changes can influence the derived topography (see the lengthy discussion in [27]). For Triton, albedo variations in the terminator region appear to be negligible at 600–1200 m scales, but albedo variations can be resolved in some portions of cantaloupe terrains in the closest approach mosaic at ~340 m scales. PC does not work in the southern hemisphere terrains because of frequent local albedo variations and the high solar illumination over most of these terrains.

Despite the innovations employed here in using PC on Triton, uncertainties in photometric properties are large enough to make longer-wavelength topographic comparisons (i.e., at scales of circa >100 pixels in the scene) suspect, and they are geologically implausible in most cases. To suppress these distortions and make the topography of smaller features interpretable, a 151 × 151 (~100 km) high-pass filter was run on the final PC DEM, rendering topographic information at these wavelengths and larger invalid in these products.

### 2.7. Smeared Images

The other unusual aspect of the Voyager Triton imaging library is that 8 of the 10 images in the closest approach mosaic were smeared by a few pixels due to camera drift during the exposures. Smearing will degrade or corrupt both SG and PC mapping due to the lateral blurring of features. We use desmeared images produced by T. Stryk (Figure 7a,b), which significantly improves stereo image viewing and improves the signal-to-noise of the resulting DEMs from these stereo pairs. These images allow us to use the entire VTERM CA mosaic for both SG and PC. The global mosaic uses the desmeared images, as does the SG_DEM and PC of the VTERM CA high resolution mosaic. Every effort was made to ensure that the photometric properties of the surface were retained as much as possible, but the desmearing process likely introduces unknown photometric deviations of small, high-contrast features, so slope values may have additional error.

For most small-scale feature measurements, we use the PC_DEM and SG-PC_DEM data derived from the relatively unsmeared 1.3–0.6 km/pix global mosaic data.

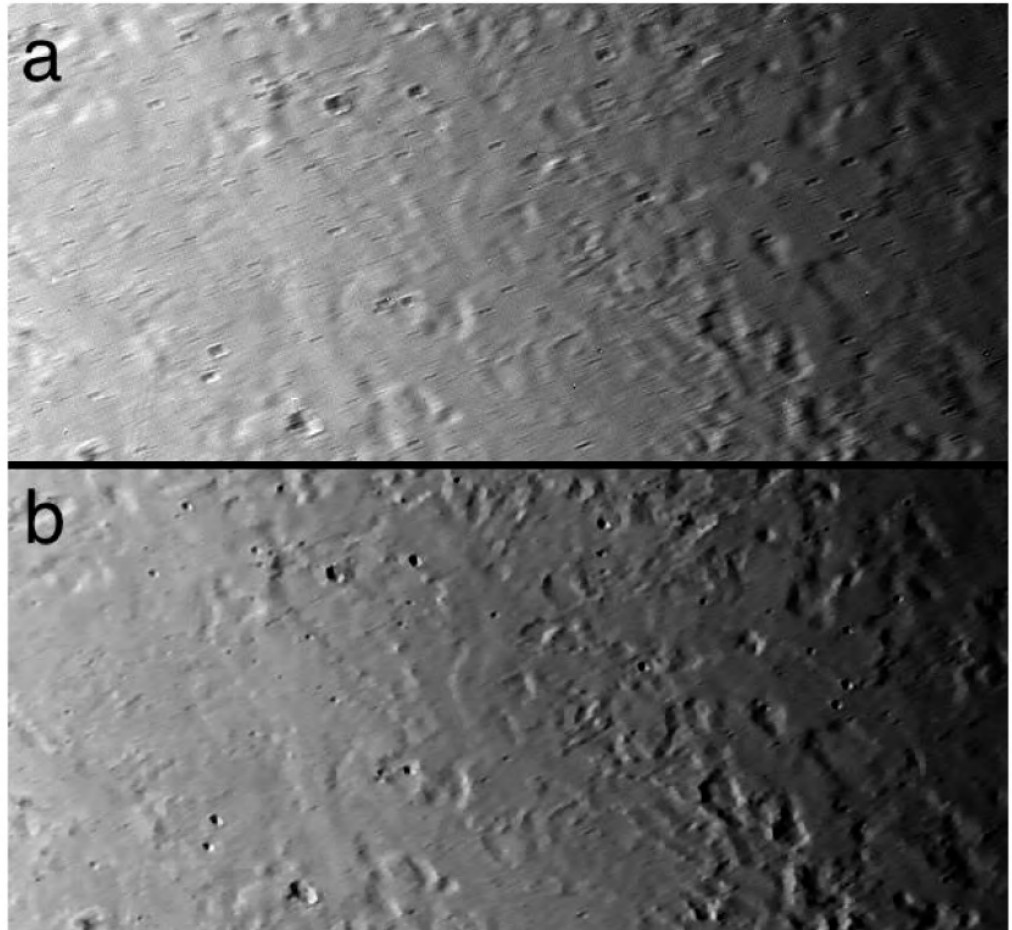

**Figure 7.** (**a,b**). Original (**top**) and desmeared (**bottom**) versions of Voyager 2 Triton image c1139607.img. Desmearing by T Stryk.

*2.8. Stereo-Controlled Photoclinometry (SG-PC)*

The SG and PC techniques have complementary advantages and disadvantages. In the area covered by the VTERM CA mosaic, both SG and PC are used and these can be combined to produce an integrated DEM of the mosaic that resolves all features at the pixel scale (Figure 8). Relief across stereo DEMs is fixed with respect to the image plane and therefore reliable at all wavelengths across the scene (Voyager reseaux ambiguities notwithstanding). However, the final topographic footprints (or pixels) in stereo DEMs are a factor of 3 to 5 larger than that of the worst resolution image in the stereo pair. PC-based DEMs have horizontal resolutions equivalent to the input high-resolution mapping mosaics but suffer from increasing uncertainties in relative elevation values as the length scale increases due to the accumulation of unresolved errors induced by the unknowable albedo variations.

Across most of the VTERM CA mosaic, stereo and low-sun images coincide, allowing us to produce SG and PC DEMs of the same area. In such cases, it is possible to use the SG-based DEM ("SG_DEM") to control the PC-based DEM ("PC_DEM;" Figures 3 and 6), effectively eliminating the suspect long wavelength information inherent to it. To merge these two types of data, we employ a set of digital filters that suppresses the long-wavelength characteristics of our PC_DEM, while retaining its short-wavelength pixel-scale characteristics. This DEM is then remapped to the SG_DEM, thus combining the high-quality high-frequency information of the PC data with the horizontal control of the stereo data.

Of course, small-scale ambiguities in the PC models can also be preserved, such as when small dark spots in the bottoms of craters or troughs are interpreted as shading and create local topographic distortions (e.g., [27]). We have discovered that the merging process tends to suppress these distortions to some degree.

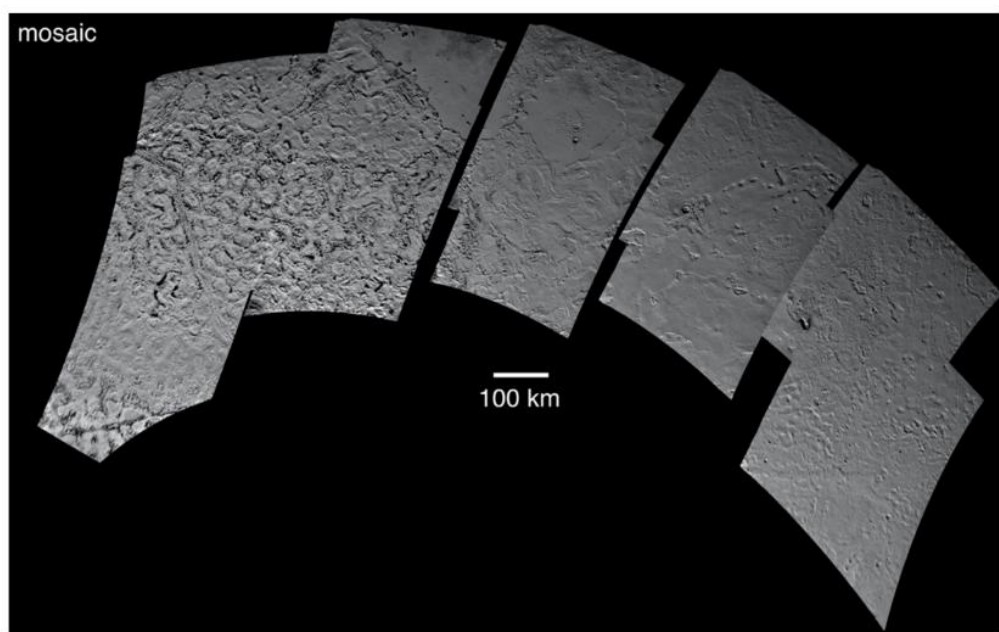

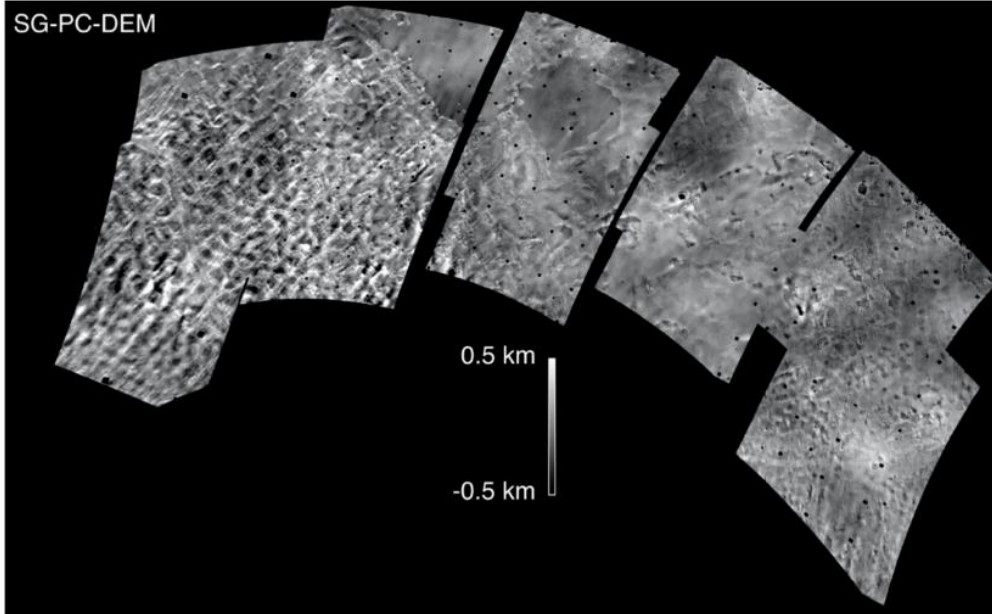

**Figure 8.** Merged SG-PC DEM corresponding with the VTERM highest resolution mosaic of Triton.

### 2.9. Using the Topographic Data

As with all PC-derived topographic data, caution is urged in interpretation. Analysis must be conducted in association with inspection of the original images and mosaics, which provide geologic context for the topography. PC_DEMs are very useful for small-scale topographic features but cannot be used reliably for features roughly 50 pixels across and larger unless additional constraints (e.g., coincident SG-DEMs) are available. Similarly, the inherent low vertical precision of the SG_DEMs relative to the surface relief of Triton, as well as the potential for subtle vertical distortion should be remembered while interpreting the data.

### 3. Results: Topographic Characteristics of Geologic Units and Structures

Prior topographic knowledge of Triton was limited to spare photoclinometric sampling [6] and several limb profiles [23]. Our new mapping results include new or updated stereogrammetric topographic mapping data across ~10% of Triton's surface, photoclinometric topographic mapping data for an additional 15% of the surface (Figures 3 and 6). These data are supplemented by the limb profiles elevation tracks [23]. Specific data products include:

1. SG_DEM corresponding to VTERM CA mosaic (Figure 3) at vertical precisions of ~300–1000 m.
2. PC_DEM covering most of the terminator region of the Voyager 2 encounter (sub-Neptune) hemisphere (Figure 6) at 1.2–0.6 km/pxl, covering ~8% of the surface, used for smaller geologic features.
3. Stereo-controlled version of the PC_DEM corresponding to the VTERM CA mosaic coverage area (Figure 8).
4. Sparse match point radius estimates extended over ~40% of the surfaces, mainly over southern terrains (Figure 8).

Except where noted (e.g., walled plains and volcanic terrains), topographic values cited here are limited to small-scale features in the PC_DEMs (Figures 6 and 8). The precision of vertical measurements in the stereogrammetry is ~250–350 m and <50 m in the and merged SG-PC products (over distances of <10 km), but to accommodate all other sources of potential error, we assume a general 20% uncertainty to topographic estimates using all data.

As any assessment of the global topographic characteristics of Triton from these data depends in part on the geologic features and terrains in the mapping area, we first examine these. Our best topographic data sets cover three major geologic terrains first described by [5] and [6]: cantaloupe terrains, smooth (volcanic) plains of Cipango Planum, and etched plains, and a variety of specific geologic landforms, including closed depressions, ridge belts, walled planitia, paterae, and catenae (Figure 9).

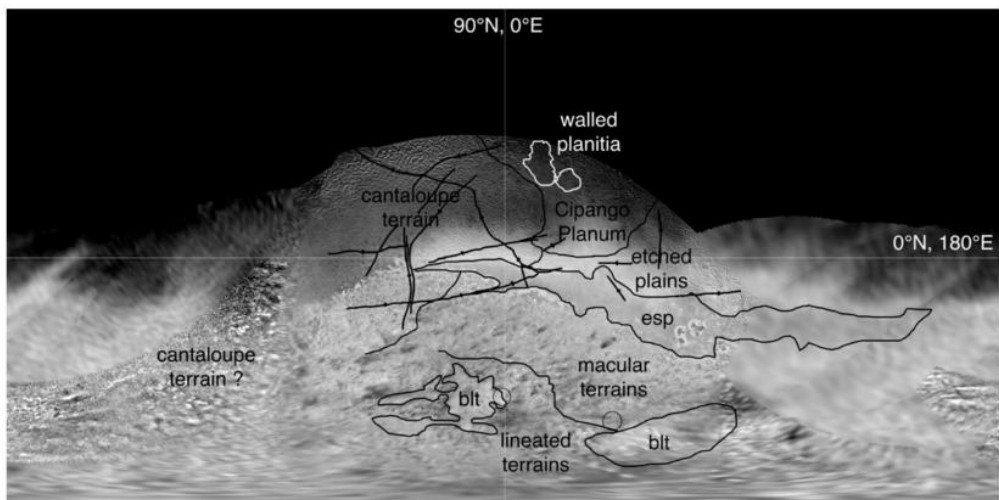

**Figure 9.** Global mosaic of Triton annotated to highlight major features discussed in the text. Narrow curvilinear markings are approximate terrain boundaries, heavy lines are linear features such as ridges Map by P. Schenk modified from [30] and [8]. Map boundaries are notional in areas where boundaries are indistinct.

### 3.1. Cantaloupe Terrain

Cantaloupe terrains may be Triton's most unique terrains (Figure 10). Consisting of topographically closed, oval to kidney-shaped cellular depressions (or cavi/cavus) 20–40 km wide over a contiguous area of at least 1400 by 2000 km [6], these nearly crater-free ter-

rains [8] have been attributed to a compositionally-driven (or possibly thermal-driven) diapir-like crustal overturn process [9]. Modelling [31] suggests that crustal compositional layering might not be the cause of the overturn, but not all compositional scenarios were explored. Additionally, the presence of compounds like ammonia in Triton's subsurface could have a significant impact on the thermal evolution of Triton [32]. The lack of resolved compositional constraints, however, hinders modeling.

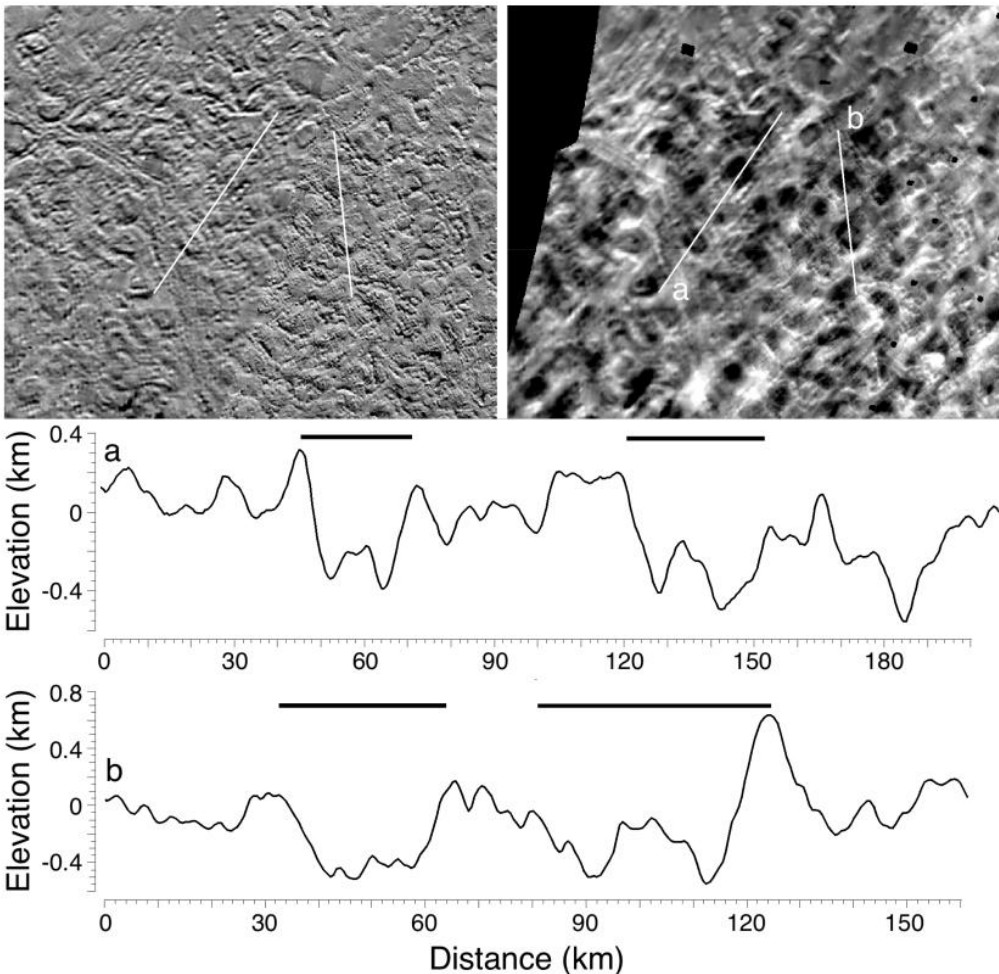

**Figure 10.** Topographic profiles across cantaloupe terrain from SG-PC-DEM in Figure 8. (**top**) Image mosaic and (**right**) DEM scene width is ~430 km. Note different horizontal scales in profiles.

The cavi are deeper than the intervening ridges, or septa, except to the north where embaying smooth plains, interpreted as (cryo)volcanic, overlap with and partially fill the cells. The cantaloupe cavi are the only individual geologic features definitively identifiable in the stereo DEMs. The unmodified cavi we can measure reliably are 200–800 m deep relative to the septa, with mean depths of ≈400 m. The PC_DEM (Figure 6) yields similar results but is subject to greater variability and longer-wavelength uncertainties because of unmapped albedo variations.

The cavi and the septa separating them are complex on sub-kilometer scales. The floors of the cells can be flat or modestly domed and can feature radial ridges, lobate scarps, and small knobs or domes (Figures 10 and 11). These smaller scale structures have estimated relief of up tens of meters, up to 100 m or so. The elevated septa between the cells have complex morphologies as well (Figure 11), including short curved ridges, some of which form tight sigmoidal curls. These ridge components are typically 100 m high, but as high as ~200 m.

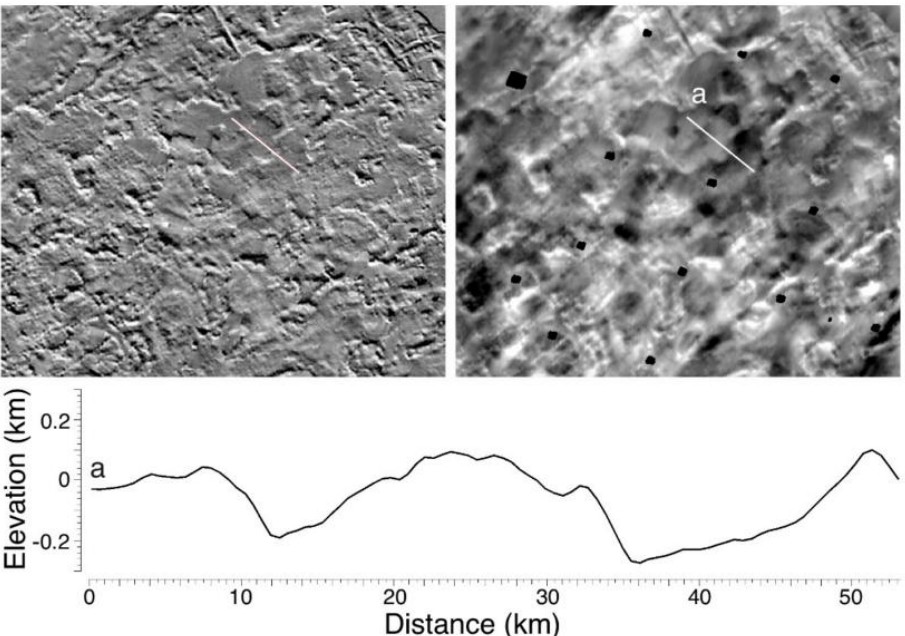

**Figure 11.** Topographic profile across scarp-bounded "layers" within cantaloupe terrain within cantaloupe terrain cavi, from SG-PC-DEM in Figure 8. (**top**) Image mosaic and (**right**) DEM scene width is ~260 km.

Many of the small-scale cavi features are likely due to processes secondary to overturn, such as flow of melted materials, folding, or sublimation erosion that have locally modified cell interiors, but resolutions are insufficient to distinguish among these alternatives. Schenk and Jackson [9] speculated that these septa could be contractional folds associated with diapiric overturn, but they could also be erosional remnants of the original surface layers.

The visual impression from shading across the image mosaic (Figure 1) is that there is little or no large-scale topographic warping across cantaloupe terrain. The SG-PC DEM (Figure 8) indicates broad undulations across the mappable areas of cantaloupe terrain of no more than ~250 m, but these cannot be verified because of the possible residual uncertainties in Voyager image distortion removal (described in the Methods section below) and associated residual ambiguities in the long wavelength fidelity of the stereo DEMs. We do, however, take the putative long wavelength relief as a maximum probable value for the range of topography across cantaloupe terrain.

*3.2. Ridges*

Although two types of linear tectonic features occur on Triton (narrow troughs and complex ridges (e.g., [6])), they are a very minor component of Triton's geologic record, compared to the extensive tectonism observed on Ganymede, Europa or Enceladus (volcanism and solid-state overturn dominating on Triton). The narrow troughs are only a few pixels across in the images and likely less than a few hundred meters deep but resemble fractures or unresolved graben and only a few have been identified.

The long curvilinear ridges that cross cantaloupe terrains and southward into the ST (Figures 8 and 12) superficially resemble double ridges observed on Europa and Enceladus in some cases [33] but differ in detail. Rather than consistently maintaining a double ridge morphology as on Europa, the Triton ridges often terminate and bifurcate and can form segments of up to five parallel ridges (Figure 12), suggesting different origins or a more complex evolution or modification. In some cases, the medial ridge is lower than the flanking ridges (Figure 12b). These ridge sets have elevations of up to ~500 m with respect to the floors of cavi depressions that they cut (Figure 12a). Whether these ridges are related to flexural or shear deformation (e.g., [33]), to deposition of material along linear

fracturs, or are erosional remnants is not clear. Some of the septa bounding adjacent cavi often appear to merge into the ridges crossing cantaloupe terrain, but beyond these sparse topographic constraints, Triton's scattered ridges remain enigmatic due to the inability to resolve cross-cutting relationships. A ridge (Figure 8) that crosses into the etched plains to the east (see Section below) is eroded or modified in appearance [6]. This ridge is 250–400 m high, suggesting that the modification process did not remove much mass.

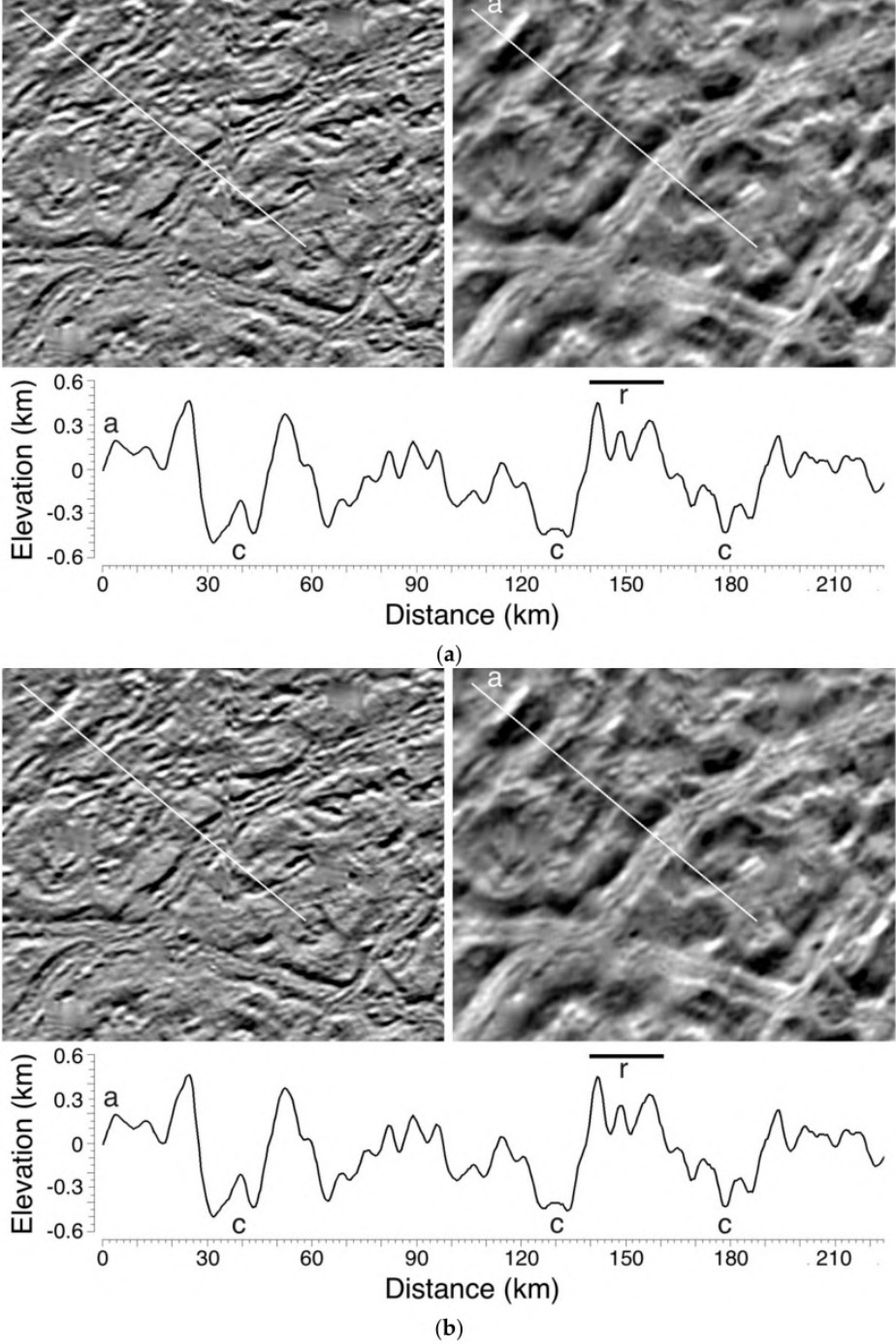

**Figure 12. (a)**Topographic profile across ridge set (r) and cavi (c) within cantaloupe terrain from SG-PC-DEM in Figure 8. (**top**) Image mosaic and (**right**) DEM scene width is ~200 km. (**b**) Topographic profile across ridge set within eastern cantaloupe terrain. (**top**) Image mosaic and (**right**) DEM scene width is ~200 km.

### 3.3. Walled Planitia

The two large walled plains Tuonela and Ruach Planitia (Figure 9) are distinct features characterized by a sinuous inward-facing scarp surrounding a low plain [6]. These scarps cut into both cantaloupe terrain and the western portions of Cipango Planum (described below). The sinuous scarps suggest scarp retreat into these terrains (e.g., [6]). Both the smaller Ruach Planitia and the southern quarter of Tuonela Planitia are barely resolved in the SG_DEM, but the SG-PC-DEM indicates depths relative to surrounding terrains of roughly 200 m (Figure 13). The DEM also suggests that the floor is subtly warped but, given potential residual artifacts from the Voyager camera distortion removal process and the poor vertical precision of the SG-DEM, the question of whether the floors of Ruach and Tuonela Planitia are level (as suggested in the images), domed, or warped as suggested by the DEM (Figure 8) must be regarded as unsettled.

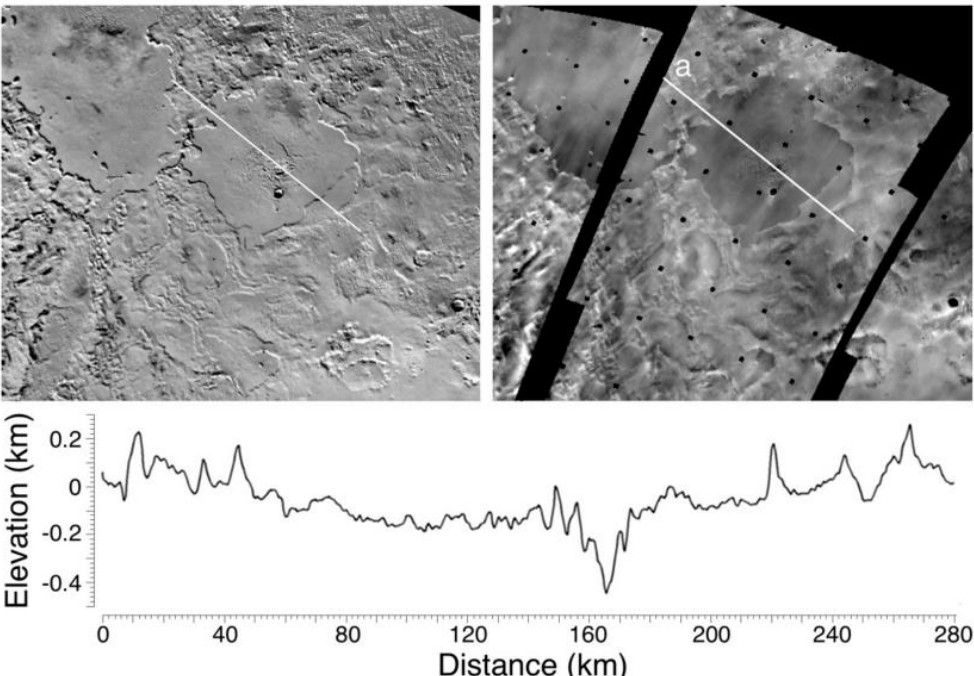

**Figure 13.** Topographic profile across the margin and central pits of the walled Ruach Planitia. Subtle undulations on the floor of the walled plains are within the stereoscopic uncertainty limits of the SG-DEM and might not be real. (**top**) Image mosaic and (**right**) DEM scene width is ~480 km from SG-PC-DEM in Figure 8.

The bounding scarps themselves are resolved topographically in the merged SG-PC_DEM (Figure 14a). As suggested by visual inspection, heights of the scarps are variable, reflecting the variable topography of the terrains these planitia cut into. Height estimates of the scarps range from ~50 m up to ~300 m, with an average of circa 150 m, consistent with the variability in relief of surrounding terrains. Maximum scarp slopes are ~15° but as they are only 1–2 pixels wide could be considerably steeper.

Textures across the planitia include marginally resolved small-scale knobs, depressions, and occasional mesa-like outliers (Figures 13 and 14) that could also be evidence of erosional scarp retreat due to sublimation, or they could be related to emplacement of endogenic materials. The largest pit in the center of Ruach Planitia is almost 10 km wide and ~350 m deep and is surrounded by smaller pits, linear troughs, and scarps <100 m deep (Figure 13). The pits in the northern corner of Tuonela Planitia are more complex, with a low dome centered within an oval depression (Figure 14b). The morphologies of pits within the planitia floors are inconsistent with impact craters, and their centralized location suggests either eruptive vent centers or sublimation of the floor materials initiated near their centers. Pits may be related to the sublimation pitting seen at highest resolution

on Pluto and Charon [34–36], but we have no constraints on the composition of Triton's planitia or any volatile that might be causing scarp retreat, if that is the origin of the scarps.

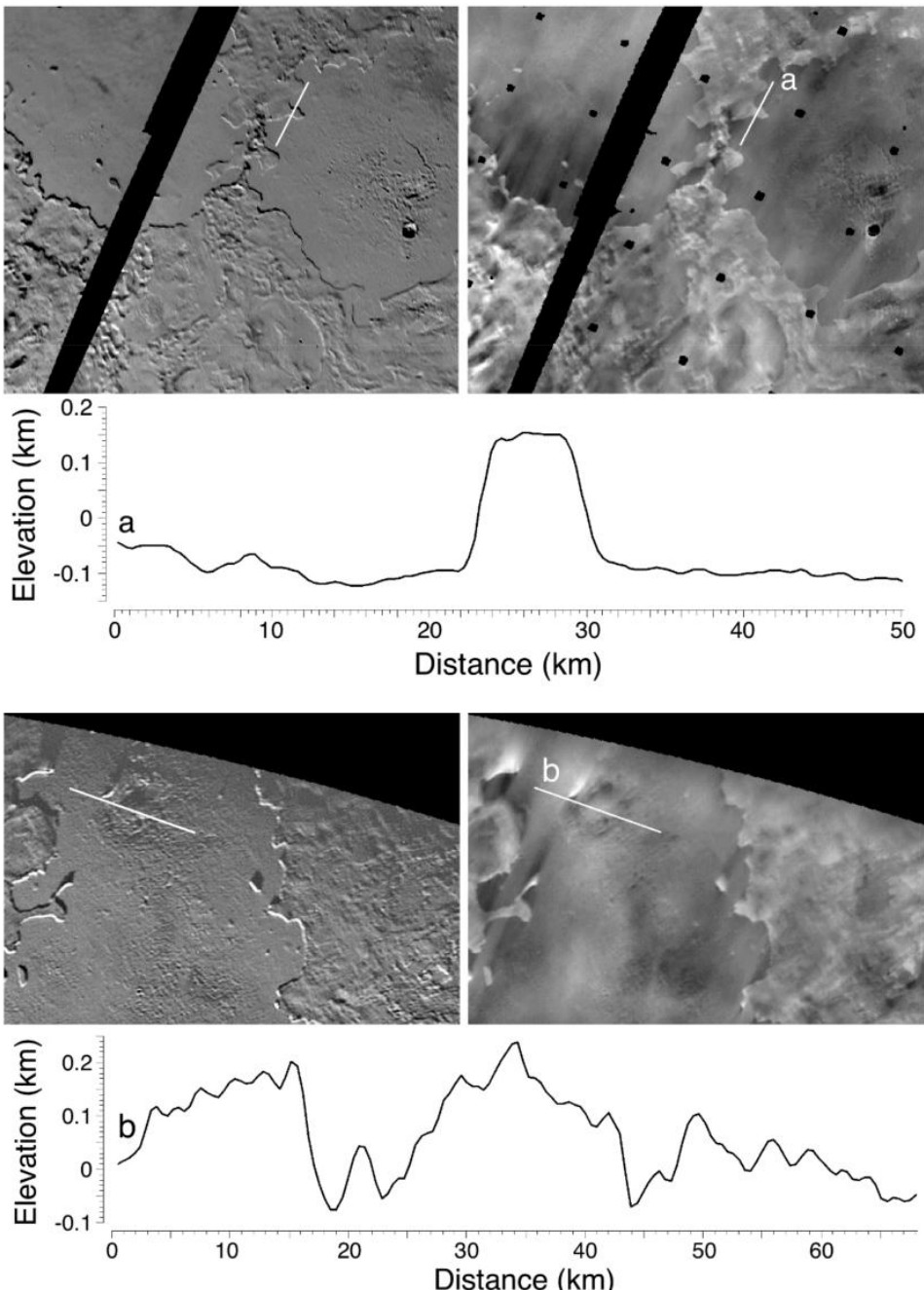

**Figure 14.** Topographic profiles across the margin of Ruach Planitia (**a**) and central pits of the Tuonela Planitia (**b**). Subtle undulations on the floor of the walled plains are within the stereoscopic uncertainty limits of the SG-DEM layer and might not be real. (**top**) Image mosaic and (**right**) DEM scene widths are ~290 km from SG-PC-DEM in Figure 8. Note different horizontal scales in profiles.

North of Ruach Planitia lies a complex knobby terrain cut by several sinuous and intersecting channels <10 km wide and ~200 m deep (Figure 15), one of which merges with northern Ruach Planitia. These are the only examples recognized on Triton and suggest that northern areas in darkness in 1989 are geologically complex. These channels are reminiscent of several eroded channels that flow into Sputnik Planitia (see Triton–Pluto Comparisons section below).

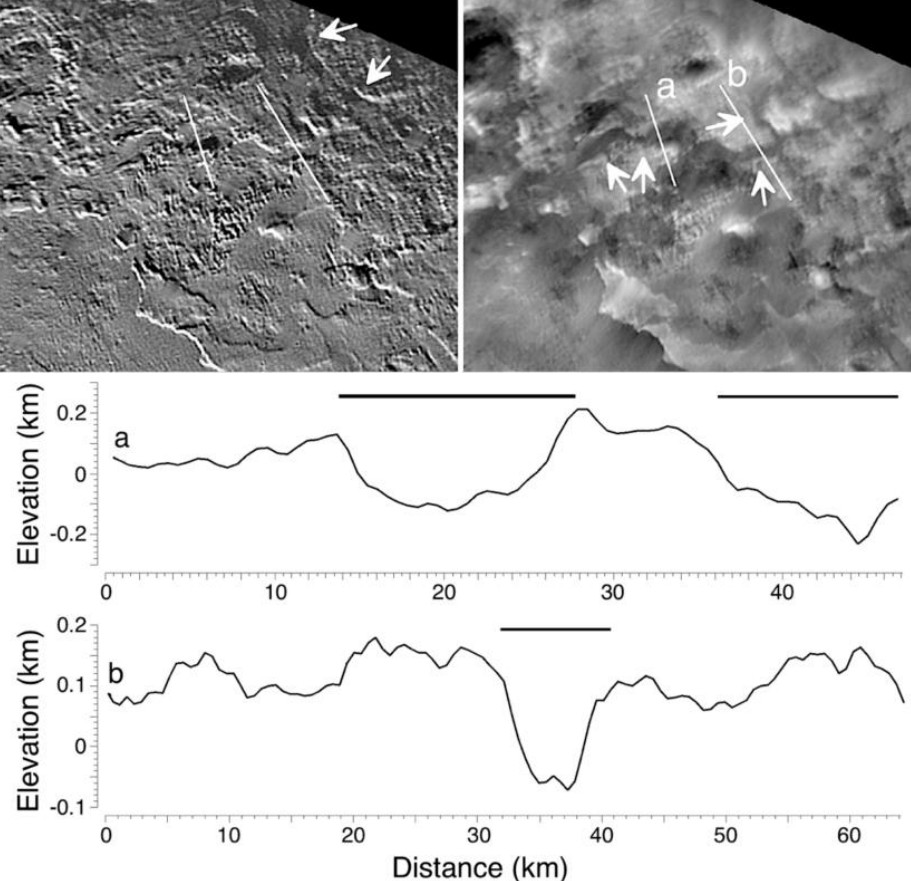

**Figure 15.** Topographic profiles across sinuous channel-like depressions north of Ruach Planitia. Note also shallow walled plains south of indicated channels. (**top**) Image mosaic and (**right**) DEM scene widths are ~150 km from SG-PC-DEM in Figure 8. Note different horizontal scales in profiles. Horizontal bars indicate extent of relevant features.

*3.4. Resurfaced (Volcanic) Plains of Cipango Planum*

The extensive plains of Cipango Planum (between 14° and 45° longitude and north of the equator) are some of the smoothest (or most featureless) on Triton and embay some areas of cantaloupe terrain (Figures 1 and 15). The plains also feature the large circular depression Leviathan Patera, as well as numerous widely distributed circular and oval pits up to ~25 km wide. These pits are both aligned in chains radiating from Leviathan Patera and as single structures in various locations in Cipango Planum (Figure 15). Cipango Planum covers an area of at least $700 \times 700$ km$^2$, but the northern boundary was in darkness. The eastern boundary is gradational, transitioning into terrains described as 'etched plains' (see below). Croft et al. [6] have described Cipango Planum as likely due to resurfacing by some form of volcanism by materials with an icy composition. Based on the morphologies, we concur with this assessment but stress that the mechanism of resurfacing in unconstrained and could involve volcanic effusion, explosive ash deposition, or both.

The ~80-km-wide Leviathan Patera appears to form the structural center of Cipango Planum (Figures 16 and 17). The northern half of the floor of this circular structure is mostly a low plain ~450 m deep and includes several rim reentrants and an irregular mesa or plateau several hundred meters high. A promontory that forms part of the northwestern section of Leviathan Patera is ~1 km high (Figure 17) and may be the highest known individual geologic feature in the topographic mapping area. The southeastern floor of Leviathan Patera is occupied by a broad rise $35 \times 53$ km across and ~400 m high (approximately level with surrounding plains), separated from the rest of the floor by a chain of linear depressions ~100–300 m deep.

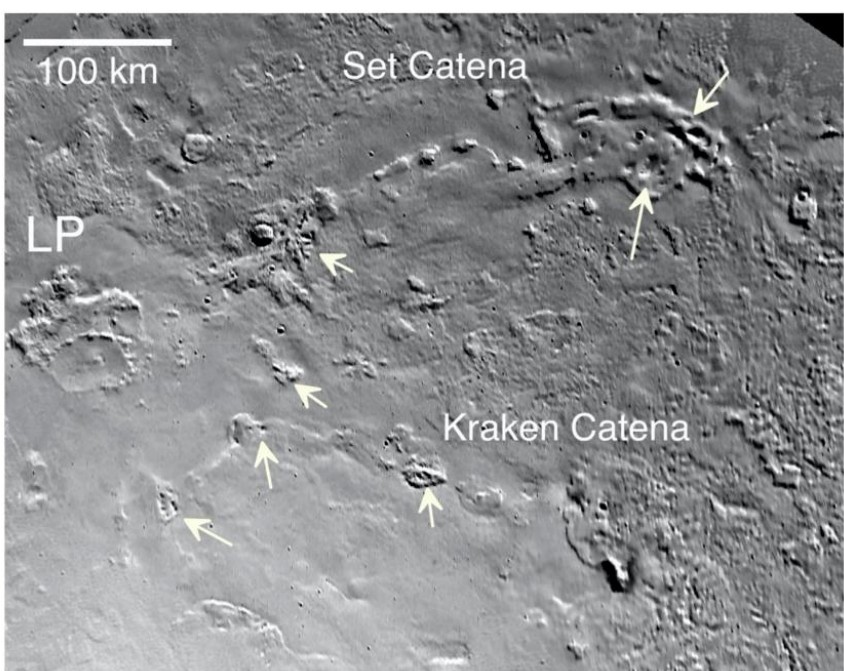

**Figure 16.** Volcanic plains of Cipango Planum showing spatial association of Leviathan Patera (LP) and Set and Kraken Catena pit chains. Arrows highlight knobs discussed in text. Scene width ~540 km.

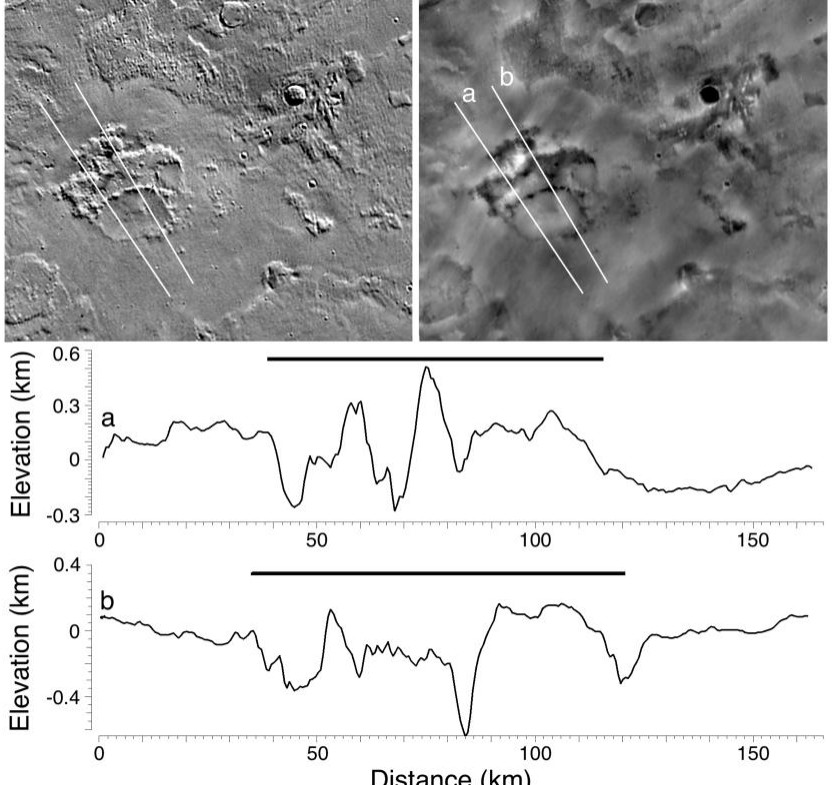

**Figure 17.** Topographic profiles across Leviathan Patera, highlighting scarp-bounded margin, broad domes in parts of the floor and high promontory on northwest patera floor. (**Top**) Image mosaic and (**right**) DEM scene widths are ~300 km, from SG-PC-DEM in Figure 8. Note different horizontal scales in profiles.

The pits across Cipango Planum occur either as linear assemblages of pits (Set and Kraken Catenae), or as a few individual, isolated pits (Figure 16). Isolated pits east and

southeast of Leviathan Patera (Figure 18) are up to 25 km wide, with the largest at 10° N, 43° E being up to 1.4 km deep, the deepest known feature in the DEM mapping areas. The largest pit might have a raised rim of possibly 100 m, but this is considered likely an artifact of local albedo variations on the PC process. The pits that form Set Catena extending northward from Leviathan Patera occur at semi-regular intervals of ~20 km (Figure 18) and are on average ~10 km wide and up to 500 m deep. These pits do not resemble impact craters and are inferred to be explosive or collapse pits [6], possibly enlarged by mass wasting. This pit chain terminates near a complex irregularly shaped partially walled depression ~80 × 100 km across and ~100–300 m deep (Figure 18), which also features several promontories up to 600 m high.

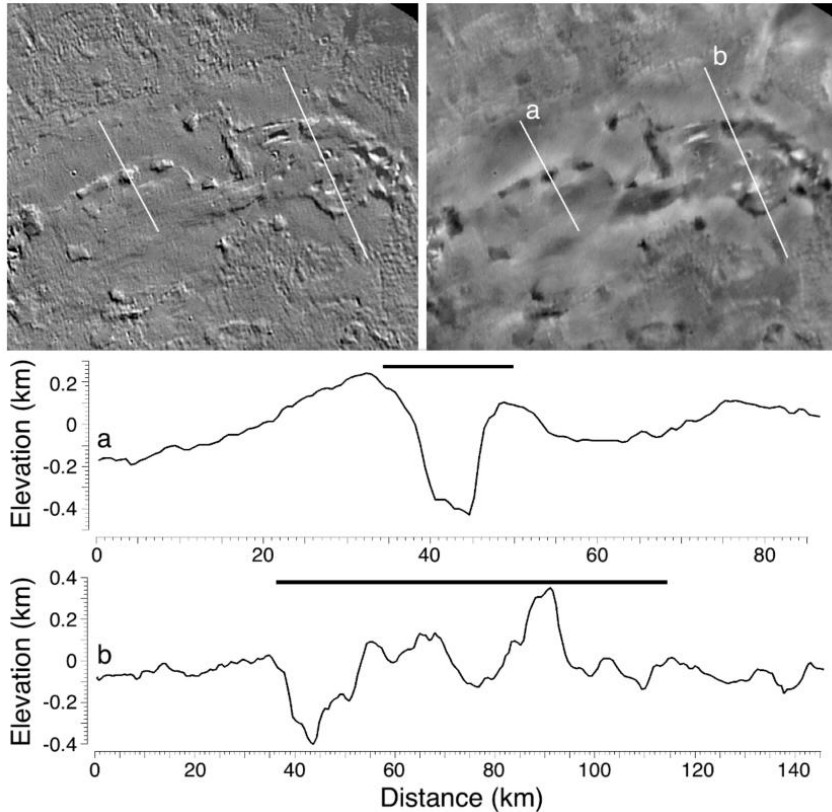

**Figure 18.** Topographic profiles across pits in Set Catena pit chain and broad shallow depression at eastern end. (**Top**) Image mosaic and (**right**) DEM scene widths are ~280 km, from SG-PC-DEM in Figure 8. Note different horizontal scales in profiles.

The curvilinear alignment of pits radiating from the central Leviathan Patera complex (Figure 16), with its low floor and interior domes, brings to mind such volcanic terrains as the Kilauea caldera and its East Rift and the Craters of the Moon rift in Idaho, albeit in its own distinctive Tritonian style. We concur with interpretations that these plains and aligned pits are a volcanic province formed from unknown ice(s) emplaced volcanically on the surface. The morphologic analogy to basaltic volcanic rift features on Earth would seem to imply similarly relatively low viscosities on Triton. With compositional mapping lacking for Triton, these materials could be ammonia- or methanol-rich (e.g., [32]), or other combinations of mobilized ice phases as fluids or fluid-solid mixtures, each of which have different viscosities (e.g., [37]). Whether the pits or pit chains are explosive or collapse features is also unresolved, but high-resolution imaging could identify whether these pits are flanked by explosive debris. As resolutions are no better than ~340 m, we do not identify flow fronts, burial, or other structures that would indicate mode of emplacement. Discrete boundaries to the smooth plains units would imply emplacement of plains forming materials was likely not predominantly explosive or "cryoclastic," if so mapped.

The broadly circular shape, low relief, and internal domes of Leviathan Patera are reminiscent of resurgent silicic calderas on Earth and could indicate that volcanism in Cipango Planum occurred in several phases: low viscosity or high effusion rate resurfacing of the plains (or pyroclastic deposition), later stage explosive eruptions or pit collapse along the pit chains and rimmed pits, and later stage extrusions of higher viscosity icy materials, or lower extrusion rates, as low eruption rates can also lead to thicker volcanic flow morphologies [38,39]. Compositional mapping and resolved imaging will, however, be required to develop a robust emplacement scenario.

The smooth plains surrounding Leviathan Patera show no evidence of significant lateral slopes in the SG-DEMs of more than ~0.1°, at least to the limits of detection. Thus, the stereo data confirm that Leviathan Patera did not form on a regional high (at least not >300 m in amplitude), as would be expected if this were a classic large shield volcano style edifice. Either Cipango Planum did not form a large volcanic pile, or if it did, the icy shell was too weak to support it and it was mostly relaxed viscously. Large-scale relaxation would suggest the presence of concentric or radial deformational features, which are not observed in Voyager data.

Several occurrences of steeper-sided promontories surrounded by low annular depressions (or 'moats') have been identified within the smooth plains (Figure 19), particularly along Kraken Catena. These depressions are morphologically reminiscent of the depressed 'moats' surrounding isolated mountains on both Charon and Ariel [20,24,34,35]. While these structures could indicate viscous flow of volcanic materials impeded by older topographic obstacles, as inferred for Charon and Ariel (see Pluto vs. Triton section below), other interpretations are possible at 350 m pixel scales, including flexural depressions due to local loading or local collapse due to magma withdrawal or some other collapse and eruptive process.

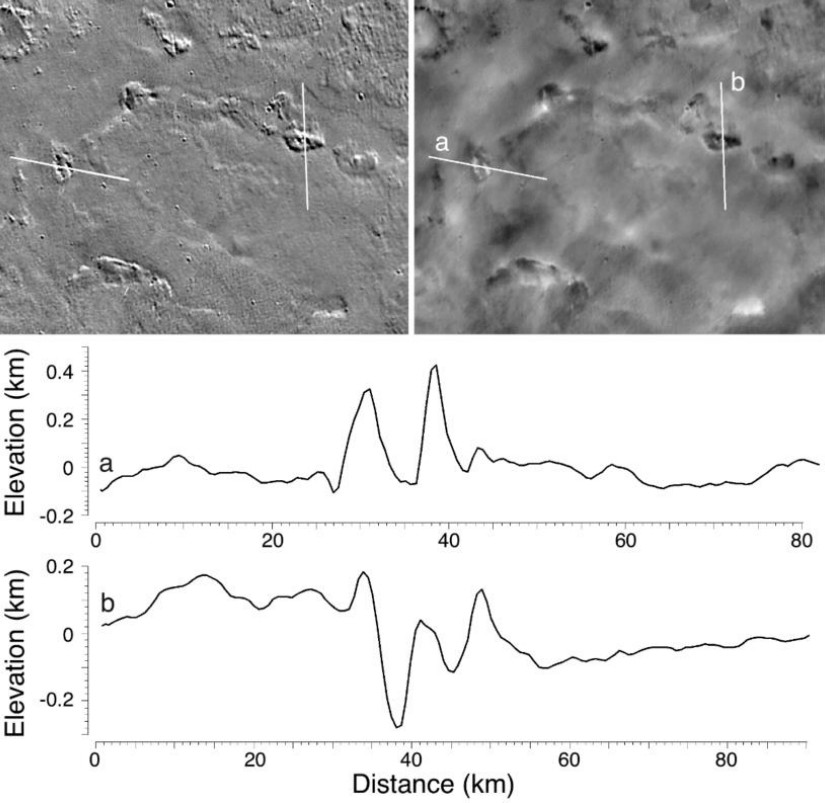

**Figure 19.** Topographic profiles across pits in Kraken Catena. These pits are noteworthy for their central knobs and apparent depressed ring "moats" between the knobs and the plains surrounding them. (**Top**) Image mosaic and (**right**) DEM scene widths are ~280 km, from SG-PC-DEM in Figure 8. Note different horizontal scales in profiles.

### 3.5. Paterae

In addition to Leviathan Patera, there are several oval closed depressions ≤100 km across (Figure 20) located predominantly along the Voyager terminator. Paterae take a variety of morphologic styles from relatively well defined and sharp-edged to more irregularly shaped walled depressions (Figure 20), but are normally distinguished by their non-circular shapes. Paterae edges are defined by scarps or ridges <200 to 500 m in height, which can vary around the circumference of the feature. The floors are either level or are domed as high as ~100 m. These patera structures lack the crenulated or scalloped margins of the walled planitia (at Voyager scales) and are distinctly non-impact related. This and their distinctive oval or kidney shapes suggest structural control by an endogenic caldera-forming process such as magmatism or collapse, perhaps analogous to uneroded silicic caldera on Earth. Whether floor doming is related to up-warping of the surface, erosional effects or volcanic construction is also unknown. Although a volcanic origin in ice is suspected, this cannot be confirmed at Voyager imaging scales, and we use the generic term 'paterae' for these structures.

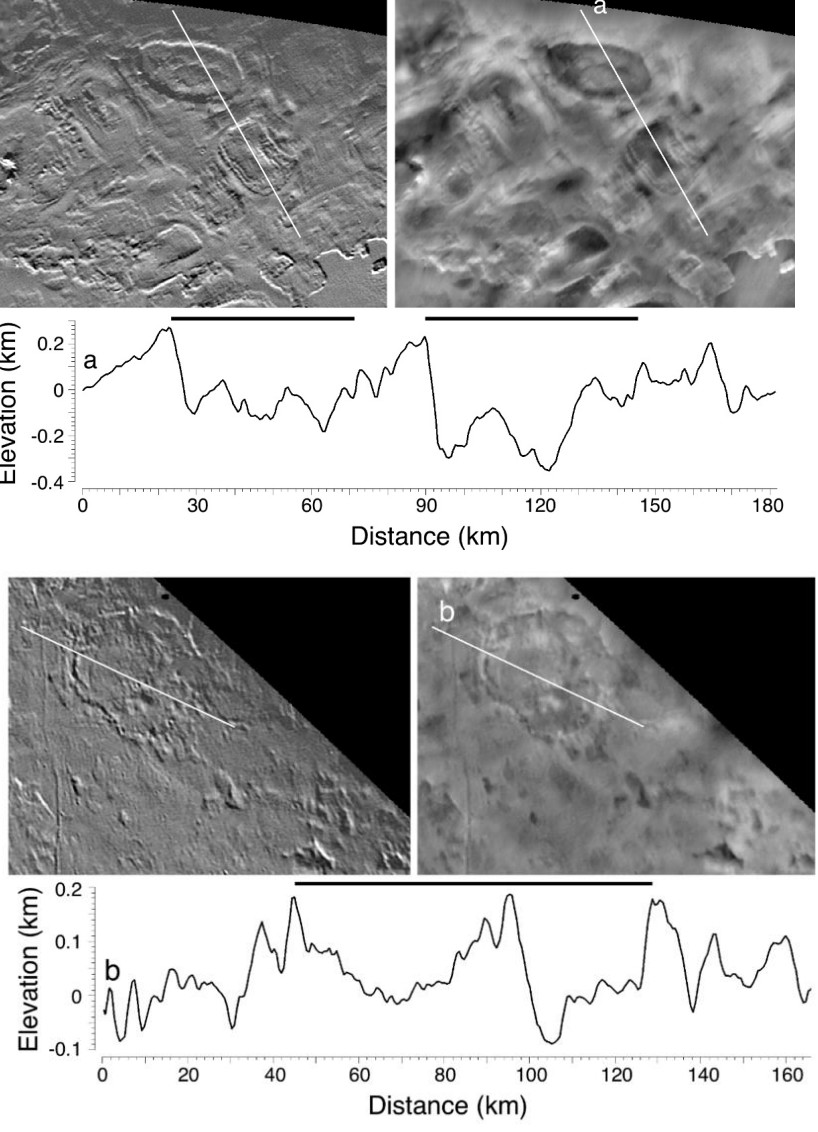

**Figure 20.** Topographic profiles of three paterae on Triton. (**Top**) Image mosaic and (**right**) DEM scene width (**a**) is ~220 km wide, northwest of Tuonela Planitia and centered at 40° N, 5° E. (**top**) Image mosaic and (**right**) DEM scene width (**b**) is ~275 km across and is centered at 14° N, 60° E, from SG-PC-DEM in Figure 8.

### 3.6. Etched Plains and Landform Degradation

Terrains east of Cipango Planum are complex in morphology, characterized by low plains marked by irregular knobs and sinuous scarps, described as likely due to mass wasting, possibly including sublimation of volatile ices [6]. In several locations, the sinuous scarps in these terrains form closed or nearly closed sigmoidal depressions (Figure 21) that are typically 5–15 km wide and 200–300 m deep. Like their larger counterparts on Pluto (e.g., Viking Planitia due northeast of Sputnik Planitia) [19], these closed depressions could be formed by sublimation and scarp retreat [40], though some of the more circular depressions could be related to the volcanism inferred for Cipango Planum to the west. More detailed comparisons to Pluto depressions are addressed in the section below. These terrains are also among the most heavily cratered on Triton, but this higher density is related to the terrains' proximity to the apex of orbital motion, which may enhance impact of planetocentric debris [8,41].

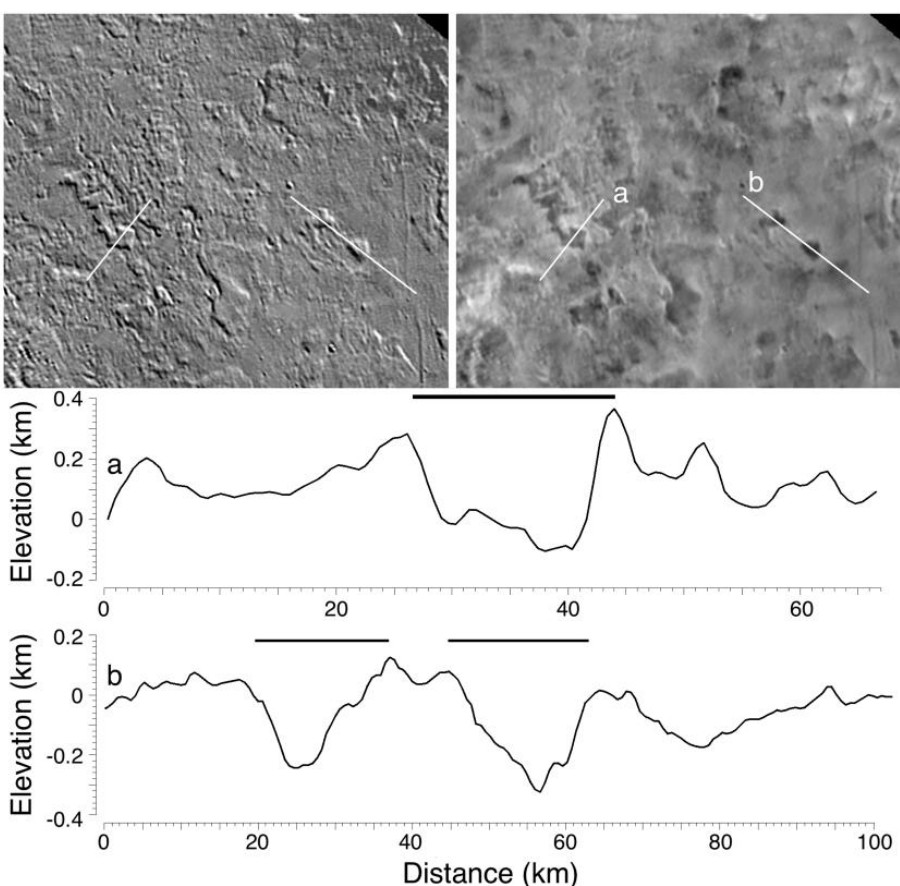

**Figure 21.** Topographic profiles across closed oval and sigmoidal troughs in the etched plains of Triton. (**Top**) Image mosaic and (**right**) DEM scene width is ~300 km and is centered at 15° N, 51° E, from SG-PC-DEM in Figure 8.

### 3.7. Southern Hemisphere Terrains

The southern hemisphere terrains (or STs) include all terrains south of the equator along a meandering boundary (Figure 22) with the above-described equatorial and northern terrains (Sections 3.1–3.5). These terrains are commonly grouped together and often (and arguably incorrectly) referred to in the literature as a "south polar cap" due to their generally higher apparent albedo than the northern terrains and collectively enigmatic characteristics. The ST are also the terrains in which the known atmospheric plumes form [11], and geologic and topographic context for their source sites and units is key to understanding their origins.

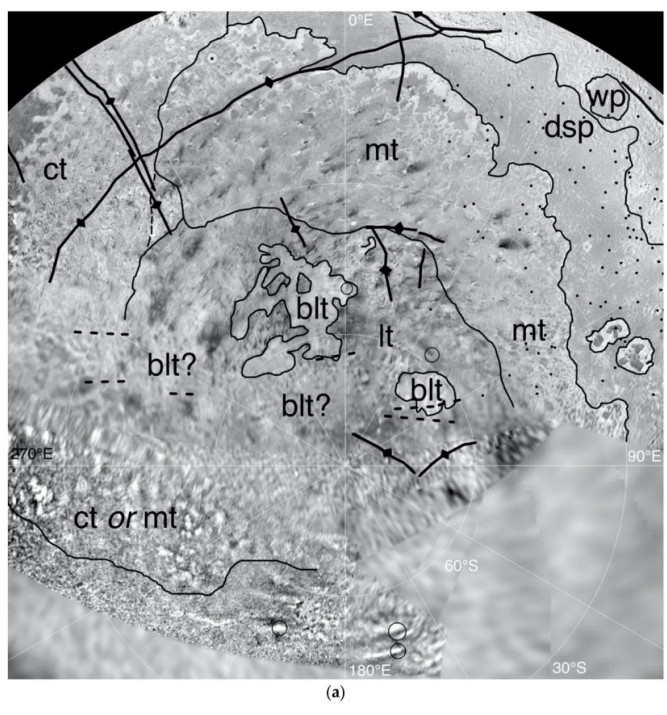

(a)

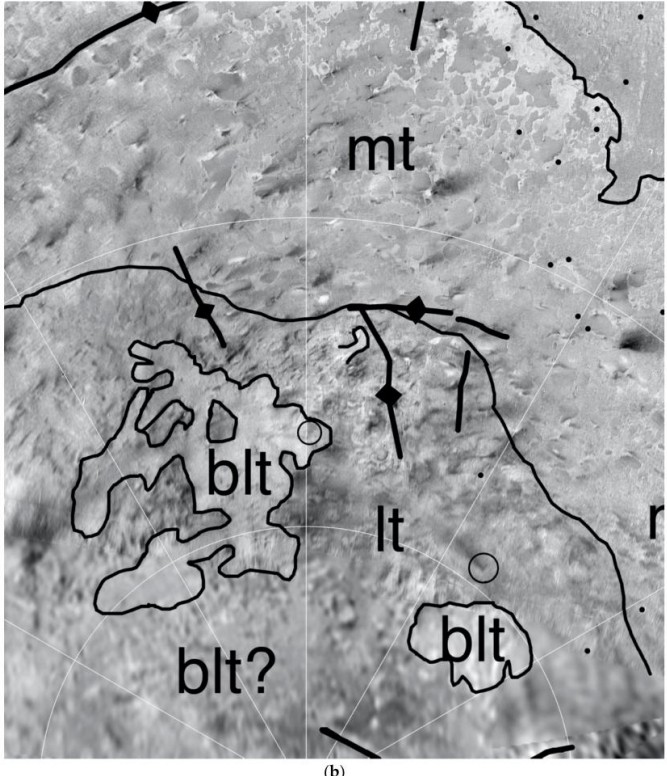

(b)

**Figure 22.** (**a**) Polar stereographic map of Triton southern hemisphere; (*mt*) macular terrain, (*lt*) lineated terrain, and (*blt*) bright lobate terrains that appear to embay lineated terrains in two large units and possible southern outliers. Note diffuse dark fan-shaped deposits in (*mt*). Terrains extending from the north include (upper right) etched plains, (*ct*) cantaloupe terrain, and (*dsp*) dark smooth plains. Open circles indicate known plume sources, heavy lines are linear features, heavy dashed lines are approximate locations of limb haze, and black dots are (not-to-scale) impact craters. Remapped from [30] and [8]. Some terrain boundaries are indistinct in the images and all boundaries are approximate, intended as a guide for the reader. Central longitude 0° E is to top. (**b**) Enlargement of central section of Figure 22a, showing southern most regions for clarity.

Because most of the ST was observed at lower resolution and in non-shadow conditions, morphologies and relief are not well expressed. The closest approach mosaic and coincident SG-PC_DEM (Figure 8) do not include the ST. The best stereo imaging over the ST uses images from the hemispheric mosaic at 1.6 to 1 km pixel scales and has relatively low parallax and estimated vertical precisions of at best ~1.25 km. Compounded by the generally low relief on Triton, this making interpretation and derivation of relief in southern terrains difficult, but not impossible.

The three published limb profiles south of the Equator (Figure 3a; Thomas, 2000) show relief of −1.5 to +2.5 km, which is greater than that observed in the northern hemisphere in the stereo DEM. Whether these correspond to actual surface features, or unresolved atmospheric plumes or haze seen in projection is not determinable from present data due to the lack of high-resolution mappable imaging, being at or beyond the edge of the higher resolution mosaics (Figures 1 and 2). Several of the images used in the global mapping mosaic also include limb segments. Many of these southern limbs (Figure 4) feature detached haze layers ~5–7 km high that can be up to half as bright as the adjacent illuminated surface and >200 pixels long in the image; at least some of these haze layers may be obliquely viewed plumes (see below for plume descriptions). The apparent heights in the limb profiles of these areas are suspect until methods of distinguishing bright haze from the optical surface in limb profiles are improved.

The only limb profiles that cross the STs away from haze layers are from the two sequential wide-angle camera images in green and violet filters taken at 3 km/pix and 99° phase angle. These cross east-west through the ST at latitudes of ≈30–35° S and mostly in the macular terrains described below. A broad 'bump' ~1 km in elevation is visible in both wide-angle limb images at ~30° S, 347° E, though this site is not geologically distinctive and could be related to the camera distortion issue. Undulations of <500 m amplitude are also evident, but as this is also approximately the effective vertical precision of the profiles, the most that can be said is that we do not see topographic deviations in macular STs of any greater magnitude than in the SG-DEM coverage to the north.

DEMs from single stereo pairs resolve the 8 km high plumes but do not resolve geologic features or topographic trends in these terrains with confidence (Figure 3c). Plume heights in the ST were remeasured using the ISIS qview stereo point measuring tool with the best stereo images available as both the plume bases and the plume fans were too indistinct in the images to measure using the SG_DEM. The tops of both plumes were confirmed to be ~7.8 and ~8.4 km high relative to apparent surface features but with uncertainties of ~1–2 km due to the poor resolution on the vent sites of the two plumes. Hili plume at least appears to originate in a circular dark spot ~3 km wide (Figure 22). One intriguing possibility to test during future Triton missions is that some of these dark spots or lineations may be sites of low velocity plume venting that does not reach the 8 km heights of the two major plumes.

To extract more useful topographic data of the ST between ~0° and 60° S, we used multiple-image control network radii solutions from the global control network solution described above to improve stereoscopic vertical accuracy in those areas relative to 2-image DEM solutions (e.g., Figure 3). The most stable radius solutions were obtained in the central longitudes of the global mosaic within Voyager frame c1139503.imq (Figure 23), the last image acquired within the ST, which provides the highest parallax with earlier frames, where we infer vertical precisions on the order of ~500 m. These data show (and are confirmed by blink comparison of the images with highest parallax) that no promontories or depressions of more than ~1–1.5 km magnitude occur anywhere within the illuminated encounter hemisphere coverage area (Figure 23). Inspection of the limb images also do not betray any large-scale relief features at these latitudes.

Southern terrains below ~10° S are not monolithic but consist of at least four distinct terrains (Figure 22): an extension of cantaloupe terrain (*ct*) west of ~340° E and north of ~20° S; macular terrains (mt) of amoeboid-shaped spots (referred to as "maculae" or "guttae") separated by brighter materials of variable albedo, generally north of ~40° S;

lineated terrains *(lt)* south of the *mt* consisting of bright and dark curvilinear and irregular surface markings; and bright lobate terrains *(blt)* with occasional faint surface markings, embedded within the lineated terrains. Due to the lower resolutions and the lack of low-Sun shading variations, the three terrains particular to southern latitudes (*mt*, *lt*, *blt*) have no obvious planetary analogs elsewhere in the Solar System, including Pluto. Those terrains closest to the south pole itself and south of ~60° S are especially enigmatic due to lower resolution, lack of shading and oblique viewing, and are not classified.

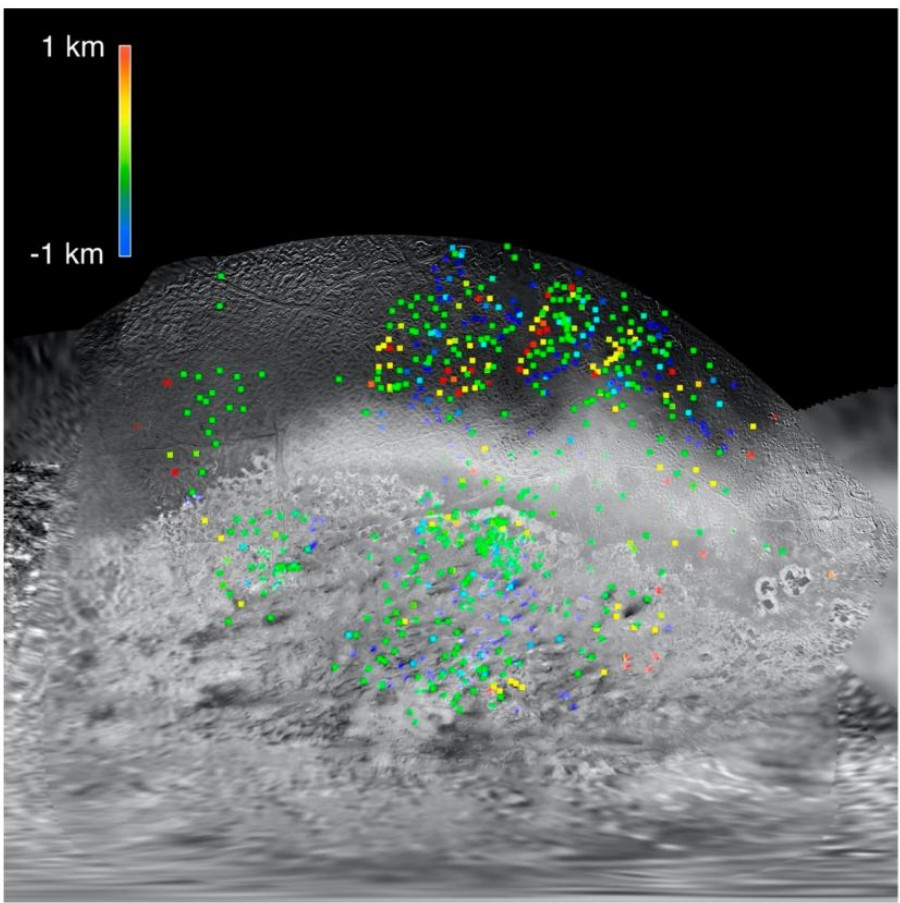

**Figure 23.** Multi-image match point radii (color spots) superposed on a global map of Triton. The area shown is from −90 to 90° E longitude.

The occurrence of cantaloupe terrains in the southern hemisphere near 10° S, 305° E (as well as its occurrence up to the western edge of the high-resolution mapping area: Figure 9) indicates that it continues as a widespread unit in the low-resolution mapping areas (Figures 22 and 24). The spotted patterns in the crescent departure images (Figures 2e, 9 and 22) are of comparable scales to cavi (but also some maculae), suggesting continuation of cantaloupe (or macular) terrains to the 50° S, 310° E area.

Although topography of the 10° S, 305° E region is lacking, the sinuous margins of the darker materials in cantaloupe terrains at this location (Figure 24) partly enclose oval brighter areas of comparable size to the depressed cavi of normal cantaloupe terrain (Figures 10, 11 and 24) and are similar to the sinuous outcroppings of intercavi septa. Apparently dark materials in this area are deposited or preserved on the higher standing septa separating cantaloupe cavi, but rather than the elevated septa being dark over an extended area, there is a gradual transition from mostly dark to most bright, suggesting a possible north-south topographic gradient in this area. Local topographic control of surface ice deposition also occurs on Pluto [19,42], except on Triton, the elevation differences are mostly <1 km rather than up to 6 km [19].

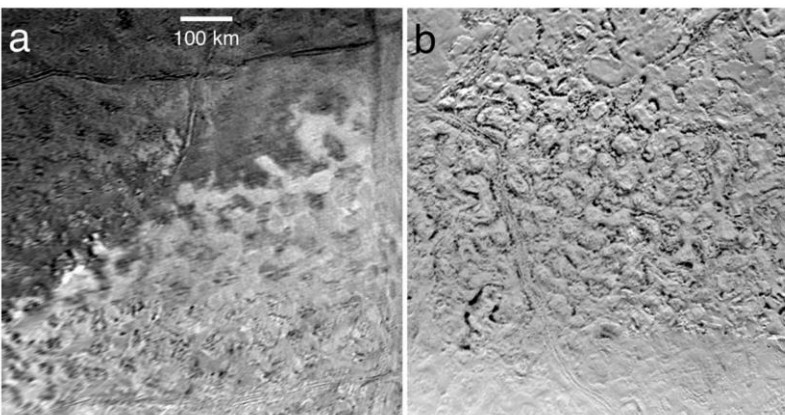

**Figure 24.** Dark surface markings south of the Equator (**a**) and high-resolution view (**b**) of the cantaloupe terrain. Darker markings in (**b**) are crests of high-standing intercave septa ridges. Images shown to same scale.

The macular terrains (Figures 22 and 25) consist of amoeboid shaped and mostly featureless darker units within a heterogenous but generally brighter unit. Most of the diffuse dark fans (e.g., [11,43]) occur in this terrain (Figure 22). The dark fans [11,43] are deposited on both darker maculae and the intervening bright materials but are usually darker on the maculae than the bright materials. The fans are likely airborne deposits given their diffuse margins and indiscriminate crossing of maculae margins, which also suggests that the maculae are not high enough to form topographic barriers to deposition. Although several of these dark fans are associated with discrete bright spots a few pixels across, neither these spots nor the numerous larger maculae on which the darker fans occur are resolved topographically. Despite this lack, we interpret the darker maculae to be higher standing than the brighter inter-maculae materials but can only infer from the lack of prominent shading or limb relief that they are a few hundred meters high at most and could be tens of meters high or less. The only instances where shading is well documented is in the large maculae at 28° S, 70° E (Figure 22), where narrow darker flanks on the eastern margins of the dark maculae indicate they are in fact elevated.

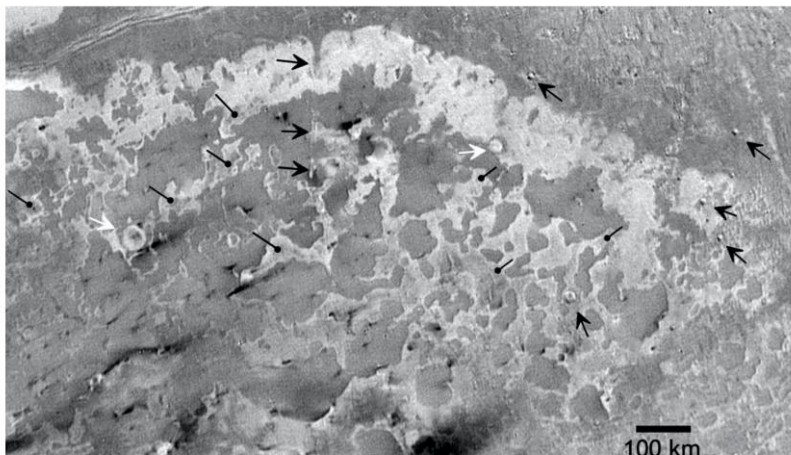

**Figure 25.** Enlargement of global cylindrical map showing macular southern terrains near the equator of Triton. The terrain consists of dark maculae units of various sizes up to ~100 km across. Between the maculae are bright materials, which are often concentrated around the edges of the patches and form diffuse contacts with a darker substrate exposed between the maculae (smaller black arrows). A few preexisting features are exposed within this dark substrate (a narrow fracture [dark arrows] and two apparently degraded craters at the left and near center [white arrows]) that may have been exhumed when the patches formed by scarp retreat. North is up.

The northern edge of the macular terrains forms a sharp but meandering boundary of brighter materials with the dark smooth plains (*dsp*; Figures 22 and 25) south of Cipango Planum and the etched plains. The nature of this terrain boundary is of some interest not only for formation and layer thicknesses but for volatile transport as the bright material might be a residual deposit from scarp retreat. The boundary scarp is not resolved topographically because of the lack of shadows and weak stereo coverage. Nonetheless, the limited useful match point radii determinations together with the lack of significant parallax shifting in the weak stereo images over this boundary indicates that the relief across the scarp is likely no more than a "few hundred meters," and is possibly <100 m. A much better understanding of this boundary, which the color mapping suggests continues another 90° or so to the east, would provide strong constraints to volatile transport models by defining the reservoir composition and thickness of N2 and other volatile ices on these terrains, and therefore their seasonal/perennial nature.

Most of the craters within the macular terrains (Figure 22) are fresh looking but several are more degraded, embayed, or infilled with the inter-macular bright materials (Figure 25). A few linear structures can also be traced well into the macular terrains (Figure 25). These craters and linea form in the inter-macular bright materials, suggesting either re-exposure of an older surface due to retreat of the maculae or incomplete burial by later-formed maculae. The brighter inter-macular materials are not always uniformly bright; in some areas, these units are brightest near the edges of the maculae, suggesting scarp retreat, control of deposition by local winds at possibly elevated maculae margins, or variable elevation on the underlying bright material substrate (Figure 25). Although poorly constrained by topography, these observations suggest a sequence in which a top-lying dark unit is being eroded by scarp retreat, exposing an underlying dark unit, with fringes of residual volatile brighter ice (Figure 28). Small spots within the maculae could be inactive venting sites.

The lineated terrains (*lt*; Figures 22 and 26) are even more enigmatic than macular terrains due to viewing conditions during the 1989 encounter, but we infer from the weak stereo images (Figure 26) and the correlation of dark albedo with elevated septa in the cantaloupe terrain noted above (Figure 24) that at least some of the irregular dark patches and linear dark markings could be the crests of higher features in this terrain if topographic control of deposition holds true here as well. This would suggest that lineated terrains may be more rugged topographically than macular terrains, though still limited to no more than ~1 km or so by the match point radii estimates, which are uncertain to within several hundred meters (Figure 23). Subtle oval arcuate structures suggest that the lineated terrains in some areas could be an outlier of cantaloupe terrain. If so, cantaloupe terrain could underly some other terrains on Triton, a speculation that can only be tested by new global mapping.

The complex albedo patterns of the lineated terrains (Figure 26) also suggest a complex interplay of surface ices of different composition. The narrow dark linear features may be elevated ridges covered by darker frosts, as on areas of cantaloup terrains (Figure 24). Some of the brighter features within the lineated terrains have lobate finger-like patterns suggestive of downslope flow of bright materials, some of which form within larger dark patches that may be elevated (based on the possible correlation of some dark material with elevation (Figure 26)). In other cases, the dark linear markings appear to be shading of scarps due to illumination (Figure 26), all of which makes interpretation of these terrains difficult in available imaging.

Some nitrogen and other volatile ices are present within the ST [17,44], but without IR-spectroscopy, it is not possible to map where they are distributed with respect to geologic units and albedo features. The bright lobate terrains (*blt*; Figure 22) may be the best candidates for large, extended, volatile ice deposits within the ST of thicknesses great enough to induce local flow. These bright lobate materials are characterized by relatively uniform albedos and lobate margins with the lineated terrains, suggesting embayment of locally low topography though no topographic data is available due to the poorly resolved

surface textures. The faint surface markings in the *blt* consist of several clusters of darker spots and lineations with unresolved relief.

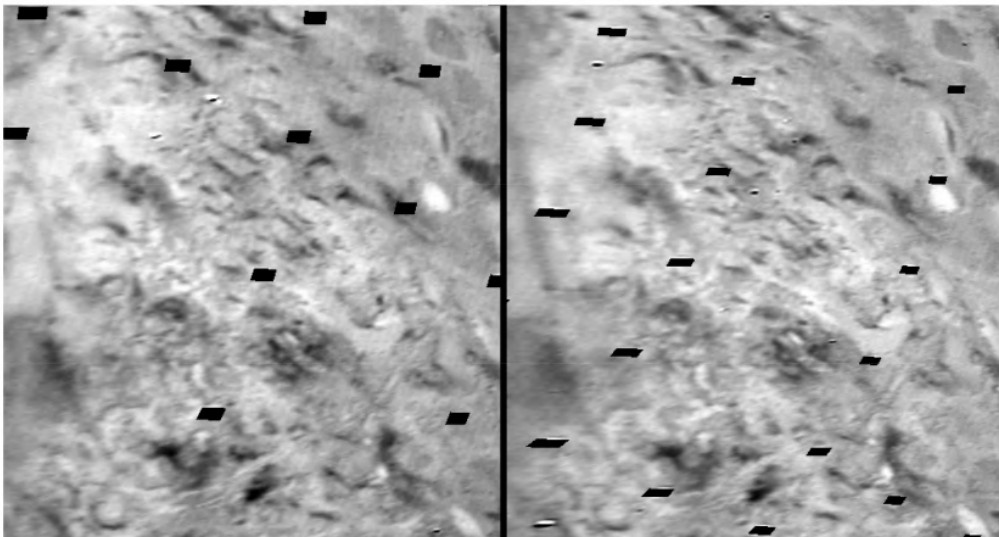

**Figure 26.** Best Voyager stereo pair of contact between lineated ST (**left**) and macular ST (**upper right**), with Hili plume and bright lobate materials at far left center. The more densely marked lineated terrains at center may be more rugged than the macular terrains at upper right with some of the darker deposits appearing to be higher standing, and narrow lobate brighter materials at lower left suggest a downslope flow. Stereo pairs such as these are difficult to interpret, however, due to the limited resolution, low solar incidence angle of <45° and low parallax. Scene width is ~400 km, centered at 44° S, 0° E; north is to right.

The best resolved example of bright lobate terrain is ≥400 km across (Figure 25 and Figure 29) and centered at 51° S, 0° E. The Mahilani plume (Figures 22 and 26) forms within circa 15 km of the edge of this deposit although the exact margin of the unit is uncertain in many areas by at least this amount. Hili plume at 58° S, 39° E originates in terrains to the east that are indistinct in the images but are also near the edge of another more poorly resolved large bright lobate unit centered at 65° S, 48° E. This deposit is ~250 km across and features several curvilinear bright lines, but it is resolved only in the 4 km whole disk images. Possible additional examples of *blt* are centered near 73° S, 343° E and 60° S, 320° E in the 4 km/pixel approach imaging. If a correlation of plume location and *blt* margins is valid, it could be consistent with a hypothesis of basal melting beneath a volatile ice unit [43] but any conclusions are frustrated by the very oblique low-resolution viewing of this area, which prevents examination and mapping of the plume site vent and local geology. Observations of other plumes with future mission observations, including topographic constraints, should allow a test of this and other hypotheses [43].

The fact that the limb hazes are not global in extent but limited in arc to a few hundred kilometers [11] suggests that they may be controlled by geology (i.e., volatile rich units or active plume vents). The projected limb location of the optically thick haze layers at circa 8 km altitude in Voyager images c1139401.imq and c1139411.imq coincide fairly closely with the location of the bright lobate unit centered at 65° S, 48° E (Figure 22a). This suggests that either the *blt* in this location is an active site of volatile loss through sublimation, producing a haze layer at altitude, or that there is an unrecognized plume source vent in this region. The haze layer in image c1139427.imq coincides with a poorly resolved area that might include bright lobate materials, pending higher resolution imaging. A haze layer in image c1139503.imq may also correlate with bright lobate materials, in an area with complex lobate margins with lineated terrains. That most limb hazes appear to be located within the lineated and bright lobate terrains could also be coincidence. The correlation of haze layers with surface geology in future mission high resolution images may prove insightful.

The absence of large circular or oval bright units on Triton similar to the >1000-km-wide 2-to-3 km deep Sputnik Planitia N2 ice sheet [19,34] suggests that topographically controlled deposits in craters >100 km scales are not preserved even on the poorly resolved hemisphere of Triton, in keeping with the inferred very young crater retention ages. Lobate margins (Figure 22) of ice deposits in the southern terrains may be confined to local-scale geologically defined depressions, as observed in other areas of Pluto (e.g., [42]).

In summary, our limited control network (CN) radii solutions (Figures 1c and 23) indicate that relief across the southern terrains of Triton is not significantly greater than ~1 km. Although observational biases are possible, this suggests that lineated and maculated southern terrains are comparable or lower in topographic amplitude than the cantaloupe and smooth and pitted terrains north of the equator. Topographic information at sub-kilometer scales for the southern terrains and especially the poorly resolved plume vent sites remains a fundamental priority for Triton.

We have identified three types of candidate mobile or volatile ice deposits in the ST. Inter-macular bright materials are variable in brightness and fringe of the darker macula (Figure 25) and may be residual deposits from scarp retreat. The localized crater filling deposits resemble on a smaller scale the altitude-controlled deposition process for volatile ices that has been described on Pluto [42] and suggests it may also work on Triton despite the generally much lower topographic relief. Narrow sinuous bright features a few 10s of kilometers in length in the lineated southern terrains (Figure 26) may be downslope ice flows. Extended volatile ice deposits on Triton do not form in large quasi-circular impact basins such as Sputnik Planitia on Pluto [34] but might be confined within several large bright lobate deposits 250–500 km across the shapes of which appear to be controlled by local topography (Figure 22). Whether any of these deposits include nitrogen, methane or another volatile ice is currently unknown, but a possible correlation with detached haze layers may indicate volatile-atmospheric exchange.

Are the STs a 'polar cap?' A polar cap can be described as substantial surface volatile materials accumulated from atmospheric deposition deposited on relatively non-volatile bedrock. As an integrated unit, the southern hemispheric terrains on Triton do not unambiguously satisfy this definition. Holler et al. [44] suggest that southern latitudes are dominated by non-volatile ices, including water ice, but allow for distributed volatile ices. The diffuse bright deposits that embay craters and the edges of the amoeboidal dark maculae (Figure 25) may be relatively thin, allowing linear structures and crater rims to be preserved. The bright lobate terrains may be thicker ice deposits but are laterally confined (Figure 22) and of unknown composition. The non-volatile "bedrock" of the STs could be of comparable age as terrains to the north, as they have comparable superposed crater spatial densities associated with the planetocentric population that may have dominated Triton (Figure 22; [8] but see [41]). Preservation of this cratering alone suggests that at least the macular terrains are longer-lived non-seasonal deposits [45]. These factors as well as the complexity and sharp delineation of features within the ST lead to the conclusion that the ST is a permanent geologic terrain, as proposed by [45], although interspersed with local and extended seasonal or slowly modifying volatile ice deposits. Hence, we do not refer to a hemispheric scale "south polar cap" but to "southern terrains" pending more robust data and assessments of these latitudes.

## 4. Triton-Pluto-Charon Contrasts

One of the major surprises of the New Horizons encounter with Pluto was that, despite the rich complexity of Pluto's surface [34], there appeared to be little obvious commonality with Triton morphology (e.g., [5,6]). The similarities in size, density, and surface composition between Triton and Pluto [15–17] and the possibility both may have been or currently are ocean worlds [3,4] allowed for the possibility that some geologic processes might have been shared by the two Kuiper belt dwarf planets [46], despite their likely very different thermal histories (Triton being captured (e.g., [47–49]) and Pluto forming in a giant collision (e.g., [50,51]). Although no compositional mapping comparable

to that at Pluto [52] was acquired by Voyager, our reassessment of Triton topography allows us to reexamine potential Triton, Pluto, and icy satellite comparisons.

Both Pluto [16,19,53] and Triton [6] exhibit terrains and features that are considered to form by icy volcanic resurfacing, but with very different morphologies. The plains and their associated pit chains and caldera-like structures in Cipango Planum (Figure 27) differ in almost every respect from the Plutonian massifs Wright and Piccard Montes, which have been interpreted as volcanic edifices [19,32,53]. Those Montes form two large circular massifs 250–300 km wide and 4–6 km high with central depressions of comparable depth [19]. The surfaces of the massifs consist of adjoining low sub-circular mounds 5–25 km across. The montes occur in the rugged depression Hyecho Planitia, which also feature several scarp bounded depressions. Cipango Planum, on the other hand, has very low relief with few mappable features except pits, pit chains, and the walled depression of Leviathan Patera, all of which have relief of no more than ~1 km. Crater densities are very low in both cases, indicating young ages [8,54].

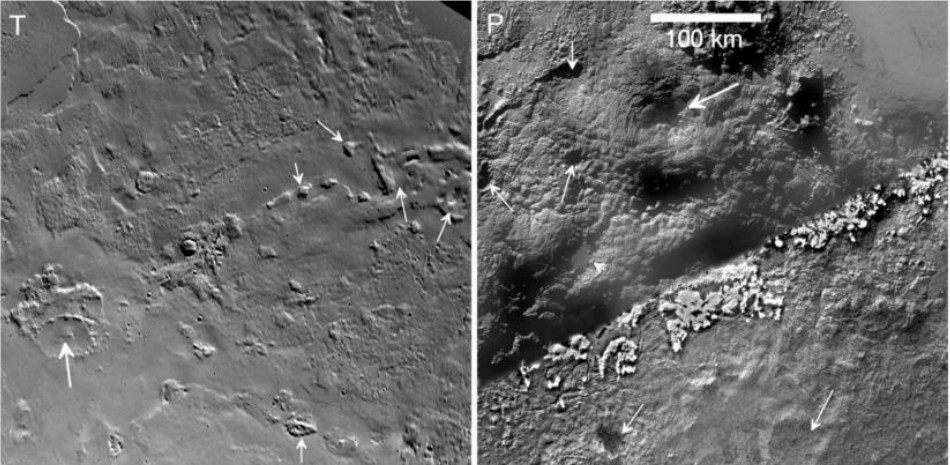

**Figure 27.** Volcanic terrains on (T) Triton and (P) Pluto. Images shown to same scale (0.350 km/pixel); scene width ~400 km. Pluto view shows Wright Mons and resurfaced plains of Hyecho Planitia.

No directly analogous structures to Leviathan Patera are observed on other icy bodies. Irregularly shaped depressions typically 20–60 km wide are found in association with smoother lanes of bright terrains on Ganymede [55] and even a pair of such features in the smooth plains of Dione [56]. While individually such examples are difficult to ascribe to volcanic origins, collectively they suggest that caldera-like collapse or explosive processes are a likely explanation for such irregular rimmed depressions.

It therefore seems likely that both Triton and Pluto experienced volcanic activity driven by internal heat but that emplacement processes and compositions of materials involved were very different. The much lower relief of the volcanic plains and constructs on Triton (<1 km (Figure 27)) suggest low viscosity ices were involved, which tend to produce lower relief deposits and can be associated with very high flow rates (e.g., [38,53,57,58]. Whether an explosive component was involved in the formation of the pit chains is not resolved. Although the physical mode of emplacement at Wright and Piccard Montes is still under debate [19,34,53], the higher relief nature of these edifices suggests either higher viscosity materials were extruded onto the surface, or higher viscosity in combination with much lower extrusion rates and volumes.

The possibility of volcanic plains forming ring depressions (or "moats") around buried topographic impediments, knobs, or ridges within the volcanic plains of Cipango Planum (Figures 19 and 28) is an intriguing comparison to the volcanic plains of Ariel and Vulcan Planitia on Charon. There, moated mountains have been interpreted as the tops of nearly subsumed foundered crustal blocks within the volcanic plains in which higher viscosity produces significant bounding scarps of 1–2 km (e.g., [20,24,35,36]). The knobs and peaks

in the Triton moat features are all <1 km high compared to the 3–5 km heights of the moated mountains and blocks on both Ariel and Charon, but there is no particular reason at present to associate the Triton knobs with foundered tilted crustal blocks. The moats on Triton are shallower by a factor 10 (100–300 m vs. 1–3 km) compared to the Charon/Ariel examples (Figure 28), again implying lower viscosity materials, though the limitations to the Triton data make interpretation more difficult.

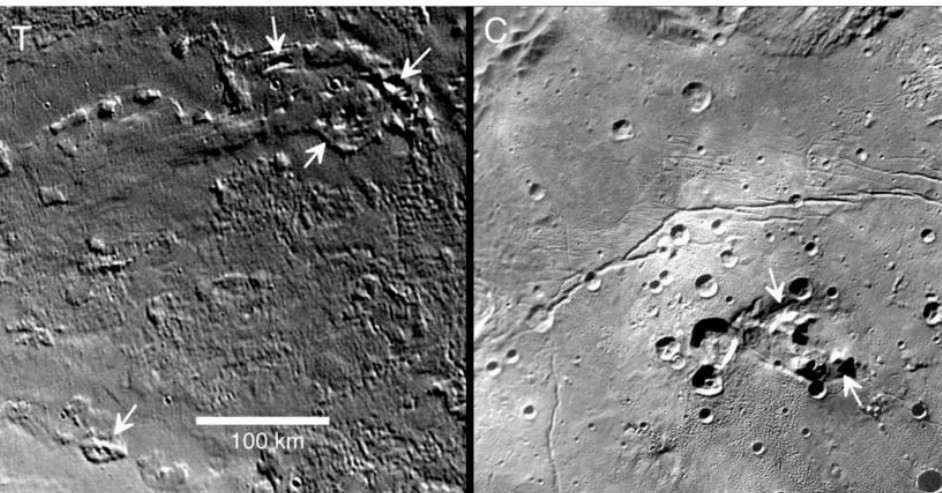

**Figure 28.** Knobs and moats around knobs and mountains on (T) Triton and (C) Charon (**right**). Knobs and possible moat on Triton are at upper right with a smaller example at lower left. Images shown to same scale (0.35 km/pixel); scene width ~300 km.

On Pluto, the closest analog in scale and outline to the two walled plains of Triton (Figure 29) appears to be Piri Planitia. Piri Planitia is partially enclosed by a scalloped and/or crenulate scarp with alcoves which is several hundred meters high [19], and the Piri Rupes scarp is characterized by alcoves and stranded mesa-like outliers suggestive of sublimational erosive retreat of a resistant scarp unit [40]. The scarps enclosing Ruach and Tuonela Planitia on Triton is more curvilinear and generally lacks erosional-style alcoves, although some stranded, barely resolved mesa-like outliers in Tuonela Planitia have been noted. Whether these differences are related to compositional differences and sublimational behavior or to Triton's youth is unknown, given the lack of compositional constraints and limited resolutions on the putative eruptive materials.

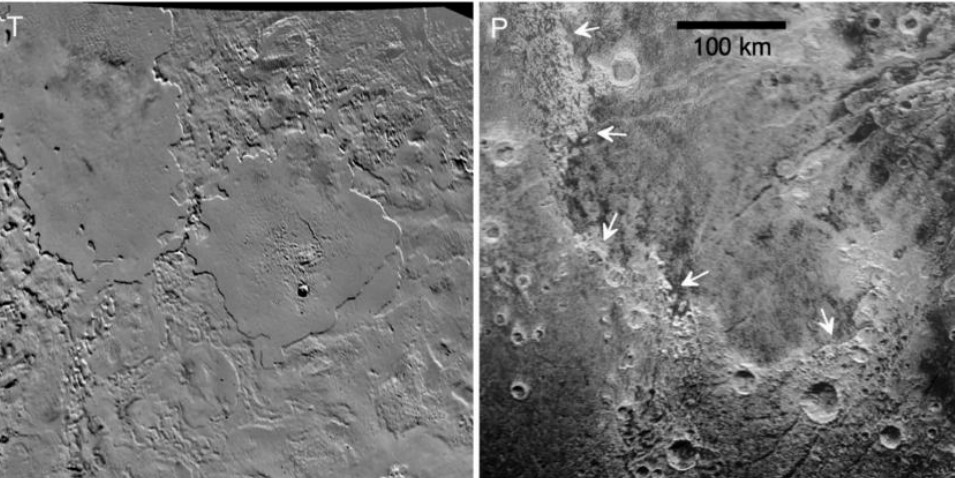

**Figure 29.** Walled plains on (T) Triton and (P) Pluto. Images shown to same scale (0.35 km/pixel) and same dimensions; scene width ~400 km.

The presence of broad sinuous channels on both Pluto and Triton (Figures 15 and 30) suggests another analog process. As both examples were not resolved at better than ~300 m, the mechanisms of formation and importance of fluid flow are unknown. Both examples open up into larger closed depressions, however, suggesting some sort of material transport occurred.

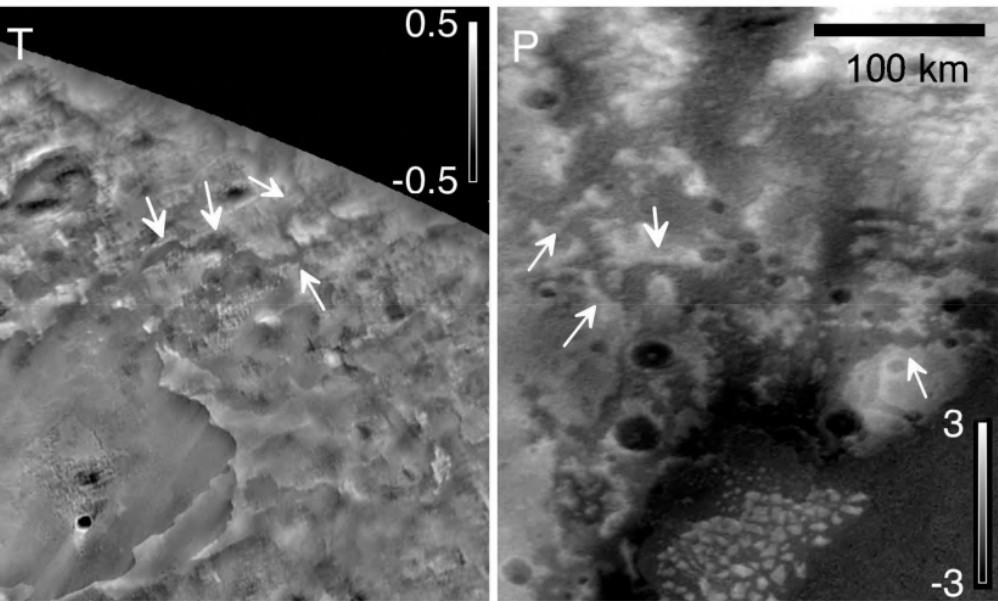

**Figure 30.** DEMs of (T) Triton and (P) Pluto showing sinuous channel-like features (arrows). Images shown to same scale (0.350 km/pixel) and scenes are ~300 km wide. Topography used in these views to highlight channel-like morphologies.

The close similarity in planform shape and dimensions of the cavi depressions in Triton's cantaloupe terrain to the cell pattern in the N2-CH4 ice sheet of Pluto's Sputnik Planitia (Figure 31) is curious. Both are characterized by adjoining and tightly packed oval-to-kidney-shaped cells roughly 20–50 km across (with some larger examples up to 75 km long on Pluto), patterns that are associated with convective solid-state overturn in extended layers with large width-to-thickness ratios [9,59,60].

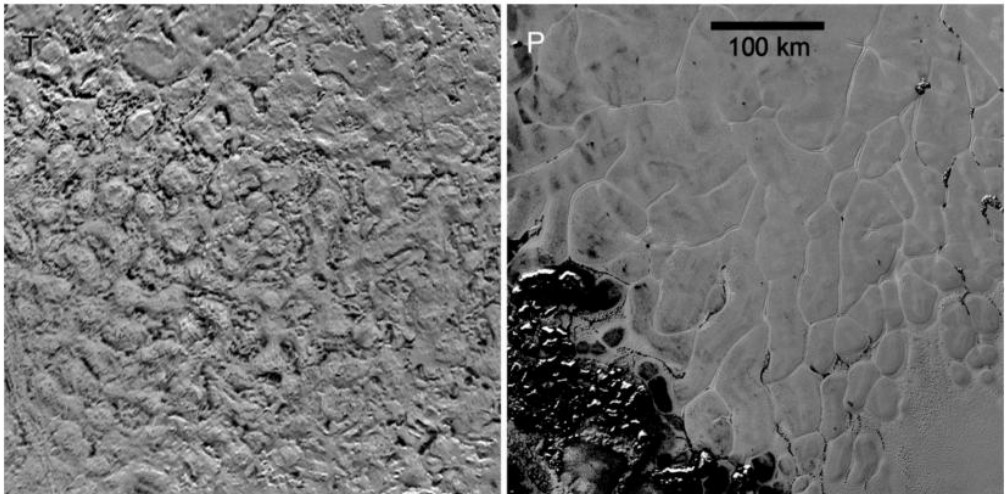

**Figure 31.** Cellular plains on (T) Triton and (P) Pluto. Images shown at same effective scale (0.35 km/pixel); scene width is ~450 km.

In the case of Pluto, the N2 (plus CH4 and CO) composition of the SP ice sheet layers is well known [52,61] and the very weak rheology of N2 ice is consistent with the very low domed relief of <200 m across most cells [19]. There are no constraints on the composition of the layers in cantaloupe terrain on Triton, which Schenk and Jackson [9] show to be associated with solid-state overturn (or diapirism). Topographically, the cells have higher relief with knobby surface textures with ≥400 m high septal ridges that separate the depressed cavi. This greater relief suggests (Figure 31) that Triton's outer layers are dominated rheologically by higher viscosity ices such as ammonia, $CO_2$, or other ices—$H_2O$ would be too rigid to convect [31]. The complex floor morphologies of cantaloupe terrain cavi (Figures 11 and 31) also suggest that a combination of extrusion and/or scarp retreat may be involved which, except for small 10–100 s of meters-scale pitting, is not observed in the Plutonian case.

In the case of Sputnik Planitia, the convective process is confined to a volatile ice layer accumulated in a topographic basin, whereas on Triton, it likely involves a substantial fraction of the upper crustal ice layers [9], possibly triggered by exceptionally high heat flow. Another difference is that septal ridges separating the cells are very narrow on Sputnik Planitia (<5 km wide and likely involving the same ice as the convecting cells), but they are wider and more complex morphologically on Triton (on the order of 10 km across) (Figure 31), further indicating that different ice(s) are involved in this process on the two dwarf planets. It is also interesting to note that cantaloupe terrain was not found elsewhere on Pluto, indicating that either Pluto's icy outer shell was too cold or thick to convect, that compositional layering requirements were not met, or such terrains were not in the New Horizons mapping areas.

The enclosed sigmoidal depressions of the eroded or "etched" terrains east of Cipango Planum on Triton (Figures 21 and 32) may be analogous to similar features in the plains north and east of Sputnik Planitia on Pluto [40] but are significantly shallower and smaller in scale. The depressions on Triton are <500 m deep and typically 5–10 km wide, whereas those on Pluto are 3–5 km deep and roughly 5–70 km in size (Figure 32). The shallower depths and smaller scales of the enclosed depressions on Triton could be due to slower erosion rates on Triton, the much younger ages of these terrains on Triton (resulting in less time for the process to proceed), or to differences in the thicknesses of the layers being sublimated.

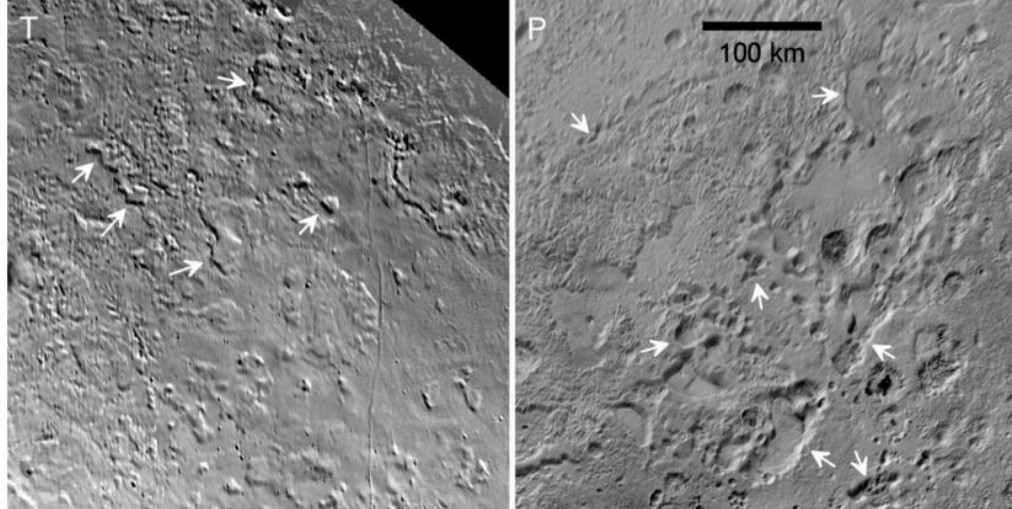

**Figure 32.** Eroded plains on (T) Triton and (P) Pluto. Images shown to same scale (0.35 km/pixel) and same dimensions; scene width is 375 km.

At least two plumes were observed to be actively erupting on Triton during the Voyager 2 encounter and explosive eruption following solid-state greenhouse heating of nitrogen-ice is one leading hypothesis for those plumes, geothermal heating being

another (e.g., [11,12,43]). Both Pluto and Triton have abundant nitrogen surface ices, but no analogous plumes were observed during the New Horizons Pluto encounter [62], a difference that is unlikely to be an observational effect. This difference could be a consequence of: (1) differences in environmental properties between Triton and Pluto (e.g., in the distribution and abundance nitrogen ice and of dark heat-absorbing materials within it); (2) different seasonal timings of the encounters (Voyager 2 when the subsolar latitude was over Triton's southern hemisphere terrains vs. New Horizons when the subsolar latitude was north of Pluto's Sputnik Planitia); or (3) another process being responsible for Triton's plumes (e.g., present-day (cryo)volcanism or nitrogen-ice melting at depth) that is not applicable at Pluto [62]. Until the mechanism(s) responsible for Triton's plumes are better understood, the significance of their lack on Pluto will also be poorly understood.

## 5. Global Topographic Characteristics of Triton

No global reference topographic baseline or geoid exists for Triton. The limb profiles from Thomas [23] are too sparse to produce a true baseline to anchor stereo DEMs as was done for Io [63]. The SG_DEM are not anchored to the center of figure. Despite this lack of an absolute reference, it is still possible to make a preliminary assessment of the global topographic characteristics of Triton in comparison to other bodies including Pluto, especially in consideration of a possible return to the Neptune system [1,2].

Given that the stereo-based topographic mapping coverage of Triton is so limited (~5% globally), it is clear that our topographic sample is potentially biased; the PC covers more area but cannot be used for regional or global elevation assessments. Although the SG-PC_DEM samples and characterizes significant areas of three major terrain types (cantaloupe terrain, volcanic plains, and etched plains), no parts of the STs are covered and (sparse) three-image solutions were required there. Although our SG_DEM is possibly corrupted by Voyager vidicon effects, any such effects are more likely to make Triton appear more topographically divergent than it actually is, meaning the topography described in the previous section should be considered a maximum. The radii DEM (Figure 23), in any case, betrays no reliable large-scale deviations of more than ~1 km amplitude if they existed in the map area and there is no conclusive evidence from shading, shadows, or limb profiles of relief substantially greater than that in the DEM. As a result, we take the DEM data as reasonably representative of the relief of those terrains covered. The limb profiles give conflicting evidence of relief in southern hemisphere terrains but are plausibly corrupted by atmospheric haze. Despite these limitations and recognizing that ours is a very incomplete survey of Triton's topographic properties, it is the best that can be currently done.

Our limited sampling of Triton topography indicates a very narrow range of relief across the three sampled terrain types (and apparently the more extensive Southern Terrains as well) of less than ±1 km at the 95% level (Figure 33). This signature is comparable to but narrower by ~50% than the global topographic signature of Enceladus, a confirmed active ocean world for which we now have global high-resolution topographic mapping (e.g., [64]). Enceladus also possesses large scale 1–1.5 km deep depressions on 100-km scales [64] that are part of this signature, which, if present on Triton, were not evident in the Voyager DEM area.

The provisional hypsogram of Triton is also 25–33% narrower than that for the ~45% of Pluto for which we have topographic data (Figure 33) [19]. Pluto's topography is strongly influenced by the 2–3 km deep Sputnik Planitia basin and the 2–5 km high plateaus of the bladed terrains, Wright and Piccard Mons volcanic edifices (see below), north-south ridge trough system and north polar dome [19], none of which are evident in the limited Triton imaging. The apparent absence of features such as bladed terrain (Moore et al., 2018) on Triton suggests that processes such as the accumulation/erosion of methane do not always occur in the same way on different Kuiper Belt objects.

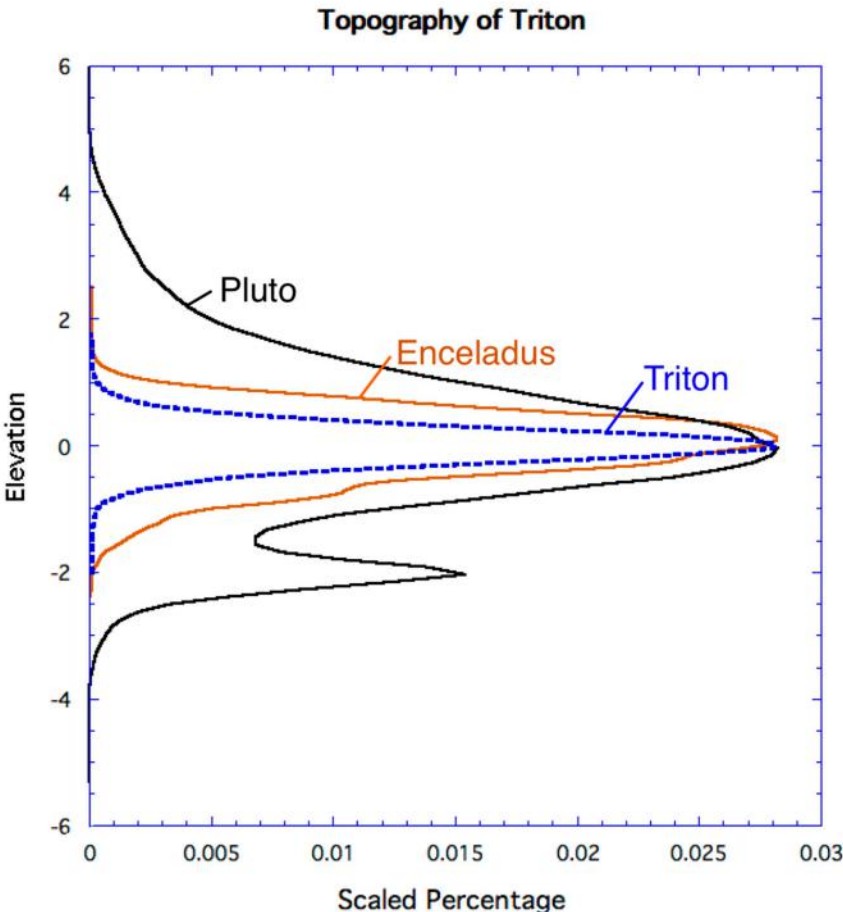

**Figure 33.** Provisional hypsogram of Triton topography based on SG_PC_DEM data. The Triton hypsogram is dashed to highlight its provisional and incomplete nature, covering only ~5% of the surface. Note that the Enceladus data [64] are global in scope and the Pluto data [19] cover ~40% of the surface, though these areas sample all known major terrain types. The Triton data do represent significant areas of three major terrain types (cantaloupe terrain, volcanic plains, and etched plains), while sparse southern hemisphere elevation data suggest similar values.

If proved accurate by future measurements, the low relief apparent for Triton (as originally indicated in the few limb profiles available from Thomas [23]) would have significant implications for Triton's ice shell evolution, heat flow and structure. The extremely young surface ages of ~100 My [7] to as little as 10 My [8] inferred from impact crater studies suggest that the topographic characteristics we observe are related to ongoing processes within a relatively warm ice shell. Further, Triton is considered a likely—though not yet confirmed—ocean world (e.g., [3,4]) and the associated elevated heat flow through the ice shell would likely have a significant impact on its topographic response as well, depending on ice shell thickness, composition, and porosity. Given the limits of the topographic data available, it would be premature to delve too deeply into the implications of these data. Nonetheless, it seems likely that Triton may possess some of the lowest topographic amplitudes of a large icy body in the outer solar system other than possibly Europa (Figure 27), or Titan [65], which are both currently poorly sampled, consistent with the likelihood of Triton being an active ocean world [4].

## 6. Discussion and Conclusions

A revised global color mosaic of Triton using updated cartographic control (with revised camera kernels) has been produced, which provides a complete, highest resolution global base from which to interpret topographic data. Stereogrammetric (SG) and photocli-

nometric (PC) DEMs of topography of Triton have also been constructed, which cover ~5% and ~8%, respectively, of the surface, and also a merged SG-PC-DEM product that also covers ~5%. Terrains covered in the stereo DEMs (Figure 3) include significant portions of the known diapiric, volcanic, and eroded plains in equatorial-northern latitudes. Match point radii solutions using three or more images provide the most useful topographic constraints on any of the southern hemisphere terrains. The PC mapping technique provides pixel-scale topography measurements over most of the terminator region of the Voyager 2 encounter (sub-Neptune) hemisphere (Figure 6) at 1.3–0.6 km/pix, and it can be used for measuring smaller geologic features but in many areas lacks stereo with which to mitigate long-wavelength ambiguities in the PC data at scales larger than ~50 pixels. Caution is always advised against over-interpreting PC results.

The limited survey of Triton's topography indicates that geologic features and terrains have very low relief and rarely exceed 1000 m in elevation, at least in the three major terrain units sampled by Voyager north of the equator. Triton may have the lowest topographic amplitude of any icy world measured to-date. Although we have sampled the three major northern terrain types, and have limited sampling of southern terrains, there may be residual uncertainties due to concerns about the geometric fidelity of the Voyager data, though these are unlikely to affect our general conclusions. No resolved topographic or geologic information is available for any of the vent geology at the known plume sites, though we verify the two plumes to be ~8 km high.

Despite the limits and uncertainties of the existing topographic data for Triton, the following observations can be made:

1.  The amplitude of geologic features (knobs, scarps, pits, ridges, etc.) is, with very few exceptions, <1 km;
2.  diapiric and volcanic terrains exhibit relief of <1 km;
3.  southern hemisphere terrains (both macular and lineated terrains) exhibit relief of <1 km, and possibly less;
4.  bright lobate terrains in the southern hemisphere south of ~45° S latitude appear to embay other terrains and may be topographically controlled volatile ices, but no reliable topography exists;
5.  bright lobate terrains may correlate with detached atmospheric hazes;
6.  no large-scale areas of mountainous terrain or deep basins of >1 km amplitude occur within the illuminated Voyager 2 encounter hemisphere;
7.  cantaloupe cells (or cavi) exhibit 300–500 m of negative relief and may be shallower where embayed by smooth volcanic materials to the north.

Although Triton differs from Pluto in many respects, several features may be similar or at least analogous. Closed sigmoidal depressions associated with scarp retreat and sublimation occur on both planets but are significantly shallower and smaller on Triton (Figures 24 and 33). The solid-state crustal convection inferred for Triton's crustal cantaloupe terrain finds no analog on Pluto, except in the much thinner nitrogen ice sheet of Sputnik Planitia. The horizontal scale of this overturn is very similar, despite the inferred compositional differences, probably a result of the lower viscosity and thinner layers involved in the Pluto case (e.g., [59,60]). The lack of such convection elsewhere on Pluto could imply that its ice shell has always been too thick and/or cold to convect, perhaps due to gas clathrate insulation [66]. Volcanic features occur on both bodies but are also much lower in relief on Triton and do not exhibit the mounded textures observed on Pluto (Figure 28). The much lower topographic relief and different morphological style (Figure 16) of Cipango Planum and Leviathan Patera on Triton, in contrast to the large volcanic edifices such as Wright or Piccard Montes on Pluto, is an indication that the viscosities associated with emplacement of Cipango Planum are lower (or effusion rates higher) than inferred for the Wright and Piccard massifs, thus implying a compositional difference on extruded materials. Of course, a Tritonian Wright Mons could have formed in the more poorly or non-imaged areas of Triton, but it seems likely that the internal thermal and compositional requirements for these structures, whatever they may be, did not occur within Triton.

The geological and topographic differences between Pluto and Triton are equally compelling, such as the apparent absence of methane-rich Plutonian bladed terrains, polar domes, large ridge-trough systems, and large impact scars on Triton. The latter two examples are very likely related to the much-longer recorded history of Pluto. The absence of bladed terrain may also be related to Triton's extreme youth, precluding time to accumulate them but also perhaps to different surface–atmosphere processes or different abundances of methane. Even when features are more similar (such as enclosed depressions), they are up to an order of magnitude shallower on Triton, further emphasizing that the geologic and thermal histories of the two planets are fundamentally different.

The general finding of very low topographic amplitudes on Triton compared to Pluto, reported before (e.g., [6,23]) and documented here in detail, is consistent with the evolution of a larger icy world with a prolonged history of elevated heat flow, which would create thermal conditions in which an icy shell cannot support and/or quickly erases large topographic loads. Future topographic mapping of Triton will require use of all available techniques, including match point radii, limb tracks (at higher resolution), photoclinometry and stereogrammetry (and a laser altimeter if practical) to elucidate the topographic properties and internal state of Triton and its ice shell.

**Author Contributions:** Conceptualization, topographic data processing and cartography by P.M.S., writing—original draft preparation, P.M.S.; writing—content review and editing, all others. All authors have read and agreed to the published version of the manuscript.

**Funding:** This research received no external funding.

**Data Availability Statement:** The Triton global color maps in CUBE and geotiff version as well as the stereo and photoclinometric Digital Elevation models and updated camera pointing kernels have been posted on the USRA Data Repository (https://repository.hou.usra.edu (accessed on 15 May 2021)).

**Acknowledgments:** Lunar and Planetary Institute contribution number 2381. The Lunar and Planetary Institute is operated by USRA under a cooperative agreement with the Science Mission Directorate of the National Aeronautics and Space Administration. We thank Peter Thomas for graciously providing the Triton limb profiles.

**Conflicts of Interest:** The authors declare no conflict of interest.

## Appendix A. Errors and Uncertainties in Topography

Error analysis in stereo SG data is relatively straightforward. The vector calculation used to derive height also reports an error, which is variable from point to point and useful as a threshold for noise removal in the DEM but has limited physical meaning. More useful indicators are the horizontal footprint and the inherent stereo vertical sensitivity or precision (VP) of each stereo pair. The horizontal resolution, or effective footprint of each topographic measurement, is a function of the size of the stereo matching patch size ($3 \times 3$ or $5 \times 5$ pixels as described above) and the resolution of the lowest resolution image in the stereo pair. The ability to detect surface elements of a given height or vertical precision (VP) is a function of the pixel scale of the lowest resolution image in the stereo pair, the characteristics of the imaging system (mainly the IFOV), and the stereo convergence angle between the two observations (as measured between the camera and the surface), using essentially the same equations as in standard aircraft photogrammetry. The vertical precision of our derived SG_DEMs is highly variable from scene to scene, ranging from ~250 m to ~1000 m.

Formal error analysis in PC mapping is complex [e.g., 27] and perhaps ultimately fruitless due to the ad hoc propagation of poorly defined uncertainties as the profile lines are integrated. PC software actually estimates slope at each pixel, not height. Although some sources of slope error can be quantified, the most important, the spatial variability of the surface albedo and our understanding of albedo and phase angle that goes into the slope calculation, is not knowable on Triton at pixel scales due to the lack of images at

various illumination conditions in the severely restricted data volume. Hence, we elect to not attempt slope or height error estimates for these maps.

The best and perhaps most effective alternative approach to understanding uncertainties in PC_DEMs is to vary the one parameter that has the greatest impact on the slope estimates, L, within reasonable limits (approximately ± 0.3) and then measuring the derived feature height along the same track when mapped using the different L factors. This approach has been demonstrated in [27]). Typically, height estimates were reasonably reliable (or at least indicative) for features at up to ~50 pixels in size. As pixel distances increased beyond this range, the reliability of relative elevations was progressively less reliable or less knowable. A second test uses shadow heights to estimate the heights of similar features to test the consistency of the PC measurements. Merging stereo with PC (SG-PC) mitigates most if not all of these uncertainties, as described above.

Thus, while elevations over short distances (<50 pixels approximately) can be considered a reasonable representation of true relief to within a few 10s of meters for typical geologic features such as ridges, scarps and knobs, longer wavelength variations in our PC_DEMs should be considered informative but not necessarily reliable, and less so with increasing scale. In the absence of independent information (SG, for example), these uncertainties cannot be characterized, and the user must use caution. With the above caveats in mind, uncontrolled PC data can provide valuable information not otherwise available for Triton. Stereo-controlled PC_DEMs retain the best high resolution and long-wavelength capabilities of each and can be considered a reasonable approximation to the true local topography of Triton at pixel scales.

As with all derived topographic data sets, users are strongly encouraged to use the monoscopic and stereo images to provide context for interpretation of the validity of the topography. An analysis of the imagery may provide clues as to whether or not there are albedo or other variations across the surface, thereby increasing confidence.

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
