# Peer review of "Triton: Topography and Geology of a Probable Ocean World with Comparison to Pluto and Charon"

_remotesensing, doi:10.3390/rs13173476_

Round 1

Reviewer 1 Report

This is a good overview and tour of Triton features using newly created topography. The comparison to other bodies is welcome also. Just minor updates as tagged in the PDF as comments.

Author Response

we thank the reviewer for the kind words.  The title and corrupted text locations have been edited as requested.  Unfortunately we do not have a direct method for tagging the match point radii values with uncertainties except to note on line 921 that they have generalized uncertainties of several hundred meters.

Reviewer 2 Report

In this manuscript, P. Schenk and co-authors derived new global digital color map products,  topography data and control networks using Voyager 2’s observations for Neptune’s moon Triton. The data analysis is clearly and very well presented and the results are carefully described and discussed in terms of geological implications. The comparison with Pluto and Charon is also very interesting. This is a paper of very high quality that should be become a reference for future studies on Triton. I only notices a few points/edits that need to be fixed before publication.

Specific points/edits: 

- l.188 - figure 3b: « Figure same as Figure 9a », it should be rather Figure 3a ?
- l. 209 : « uncertainties associated with topographic measurements » It might be useful for readers to mention here that more informations of errors and uncertainties are provided in Appendix A. 
- l. 222 : « figure 3d » it should be figure 3a ?
- l.365-378: « smeared images » Some details on the desmearing algorithm used here  should be given. It is mentioned that different desmearing  algorithms may result in different image results, how does it affect the topographic estimates ? 
- l.431 : « We assume a general 25% uncertainty to topographic estimates using all data «  Even if there is Appendix A provided more information (that I discover at the end of my reading), it might be useful to develop a little bit more here regarding the error estimates and to reference Appendix A for more details. It might be useful also to indicate on each elevation plot a estimate of the uncertainty
- Figure captions: in many  figure captions (15, 17, 18, 19, 20, 21 etc.) it is just indicated DEM without specifying if it is PC-DEM or SG-DEM
- l. 893: « Figure MAC »  figure numbering problem
- Appendix A - Table 1: Table 1 is missing. 

Author Response

We thank the reviewer for their comments.  all the Figure and Table information has been corrected as requested.  the Uncertainty issues are addressed in the Methods section but also a modestly clarified overview of the uncertainties has been edited on lines 434-455 in the introduction to the Results section.

Reviewer 3 Report

This paper examine the distribution, quality, and implications of stereo and shading derived topographic data for Triton, together with new global multicolor surface maps produced during the topography derivation. I think the paper needs to make modifications. The structure of the article needs to be reorganized and the language needs to be modified. This paper describes too many details of previous literature, and it is difficult to find the innovation of this paper.

1.The abstract must be reorganized. It should include the problems from previous research, your method to fill the gap, and the main findings and conclusions.

  1. The method part also needs to be rewritten. The beginning of the method part almost describes the data and methods of the previous literature. Should this be moved to the introduction part? The description of data and methods in this study is not clear enough, and this section needs to be reorganized.
  2. The result and discussion part should also be reorganized. The discussion lacks sufficient comparison with the previous literature. I suggest that the discussion and the results sections be written separately.
  3. What are the advantages of the three topographic data? Why use this data. This should be briefly outlined in the method.
  4. “We assume a general 25% uncertainty to topographic estimates using all data.” Why 25%?add the citation?
  5. Please add evaluation of the accuracy of the results.

Author Response

1.The abstract must be reorganized. It should include the problems from previous research, your method to fill the gap, and the main findings and conclusions.  >> the abstract opens with a synopsis of the basic problem and the tools used to address it.

  1. The method part also needs to be rewritten. The beginning of the method part almost describes the data and methods of the previous literature. Should this be moved to the introduction part? The description of data and methods in this study is not clear enough, and this section needs to be reorganized.  >> we have reorganized the beginning of the methods section lines 70-73 to remove the background material into the preceding paragraph, and start with current methods.  Previous literature is embedded through out and cannot be easily extracted.  we did provide a summary of that at the very beginning of the Description section starting lines 433.
  2. The result and discussion part should also be reorganized. The discussion lacks sufficient comparison with the previous literature. I suggest that the discussion and the results sections be written separately.  >> similarly, the comparison with previous literature is sparse because topography hasn't been used geologically very much on Triton but that is embedded throughout and not easily extracted.  We do briefly cite these works at the beginning of the Results section starting lines 434, which also very clearly lays out what new products were produced here.
  3. What are the advantages of the three topographic data? Why use this data. This should be briefly outlined in the method.  >> a short 1-sentnc e description of the main advantages has been added at the beginning of each of these sections, i.e., at lines 226-228, 307-312, 319-320.
  4. “We assume a general 25% uncertainty to topographic estimates using all data.” Why 25%?add the citation?>> the discussion of the uncertainties has been enlarged on lines 448-450 but is described in more detail in the Methods section at the end.  there is also ample discussion of the theory behind these in more detail in the cited references.  Due to the length of the article it was decided not to go into the theoretical aspects in detail here.  The 20% uncertain is an assessment based on prior experience and is conservative.  
  5. Please add evaluation of the accuracy of the results.  >> this has been specified on lines 448-450 and in the Methods section at the end.

Round 2

Reviewer 3 Report

The manuscript improves a lot after revision according to the comments, I would like to suggest to publish it.